# Bending forces and nucleotide state jointly regulate F-actin structure

Matthew J. Reynolds[1], Carla Hachicho[1], Ayala G. Carl[1,2], Rui Gong[1] & Gregory M. Alushin[1✉]

ATP-hydrolysis-coupled actin polymerization is a fundamental mechanism of cellular force generation[1–3]. In turn, force[4,5] and actin filament (F-actin) nucleotide state[6] regulate actin dynamics by tuning F-actin's engagement of actin-binding proteins through mechanisms that are unclear. Here we show that the nucleotide state of actin modulates F-actin structural transitions evoked by bending forces. Cryo-electron microscopy structures of ADP–F-actin and ADP-$P_i$–F-actin with sufficient resolution to visualize bound solvent reveal intersubunit interfaces bridged by water molecules that could mediate filament lattice flexibility. Despite extensive ordered solvent differences in the nucleotide cleft, these structures feature nearly identical lattices and essentially indistinguishable protein backbone conformations that are unlikely to be discriminable by actin-binding proteins. We next introduce a machine-learning-enabled pipeline for reconstructing bent filaments, enabling us to visualize both continuous structural variability and side-chain-level detail. Bent F-actin structures reveal rearrangements at intersubunit interfaces characterized by substantial alterations of helical twist and deformations in individual protomers, transitions that are distinct in ADP–F-actin and ADP-$P_i$–F-actin. This suggests that phosphate rigidifies actin subunits to alter the bending structural landscape of F-actin. As bending forces evoke nucleotide-state dependent conformational transitions of sufficient magnitude to be detected by actin-binding proteins, we propose that actin nucleotide state can serve as a co-regulator of F-actin mechanical regulation.

Actin polymerization powers fundamental cellular processes, including cell migration, organelle dynamics and endocytosis[1]. The propulsive assembly of branched actin networks is coupled to nucleotide consumption by globular actin (G-actin) subunits, which hydrolyse bound ATP concomitant with their incorporation into growing F-actin[1]. An elastic Brownian rachet model, wherein bending filaments push against membranes as they polymerize[2,3], quantitatively explains the conversion of this chemical energy into mechanical work. Consistently, electron microscopy has resolved abundant bent filaments featuring radii of curvature in the hundreds of nanometres within polymerizing actin networks adjacent to surfaces in vitro[7] and in cells[8–10].

After incorporation into F-actin, subunits rapidly hydrolyse bound ATP in seconds. This produces metastable ADP-$P_i$–F-actin that persists for minutes before phosphate release, resulting in long-lived ADP–F-actin[1]. Retrograde flow simultaneously pushes aged filaments away from the membrane[11], producing a spatial gradient of F-actin nucleotide states—a biochemical marker of filament age[6]. Both actin nucleotide state and mechanical forces[1,4,12,13] modulate interactions between F-actin and key actin-binding proteins (ABPs) that govern actin network dynamics. The F-actin-depolymerizing protein cofilin preferentially binds to and severs ADP–F-actin[14,15], facilitating the selective disassembly of membrane distal, aged F-actin. Cofilin and the branched F-actin nucleator ARP2/3 also feature mechanically tuned[5] F-actin-binding activity[16–18], specifically responding to filament bending[17,19]. This may contribute to force sensitivity in branched F-actin networks, which display modified interfilament geometry[20,21] and enhanced force production[20] in the presence of resistive loads. Several other ABPs detect the nucleotide state of F-actin[13,22] and force across individual filaments[23–25] through mechanisms that are unclear.

F-actin polymerization is coupled to the G-to-F transition, a substantial flattening of the actin subunit that renders its nucleotide cleft active site competent for hydrolysis[26]. By contrast, cryo-electron microscopy (cryo-EM) studies at a resolution of 3–4 Å found only modest nucleotide-state-dependent changes in F-actin, reporting either localized rearrangements[27,28] in actin's flexible D-loop[29–31] or nearly identical backbone conformations[32]. However, binding by ABPs can evoke substantial rearrangements, notably cofilin, which stabilizes an undertwisted F-actin lattice[33,34]. This discrepancy challenges a traditional allosteric model for F-actin nucleotide state discrimination by ABPs, which cannot directly access actin's buried nucleotide cleft. The actin nucleotide state could also modulate F-actin's deformability and corresponding capacity for rearrangements mediating ABP binding[13,15], consistent with micrometre-scale persistence length measurements showing that ADP-$P_i$–F-actin is stiffer than ADP–F-actin[35]. Molecular dynamics simulations support mechanical regulation of F-actin structure[36], with models abstracted to the subunit level predicting coupling between filament bending and helical lattice twist modulation

[1]Laboratory of Structural Biophysics and Mechanobiology, The Rockefeller University, New York, NY, USA. [2]Tri-Institutional Program in Chemical Biology, The Rockefeller University, New York, NY, USA. ✉e-mail: galushin@rockefeller.edu

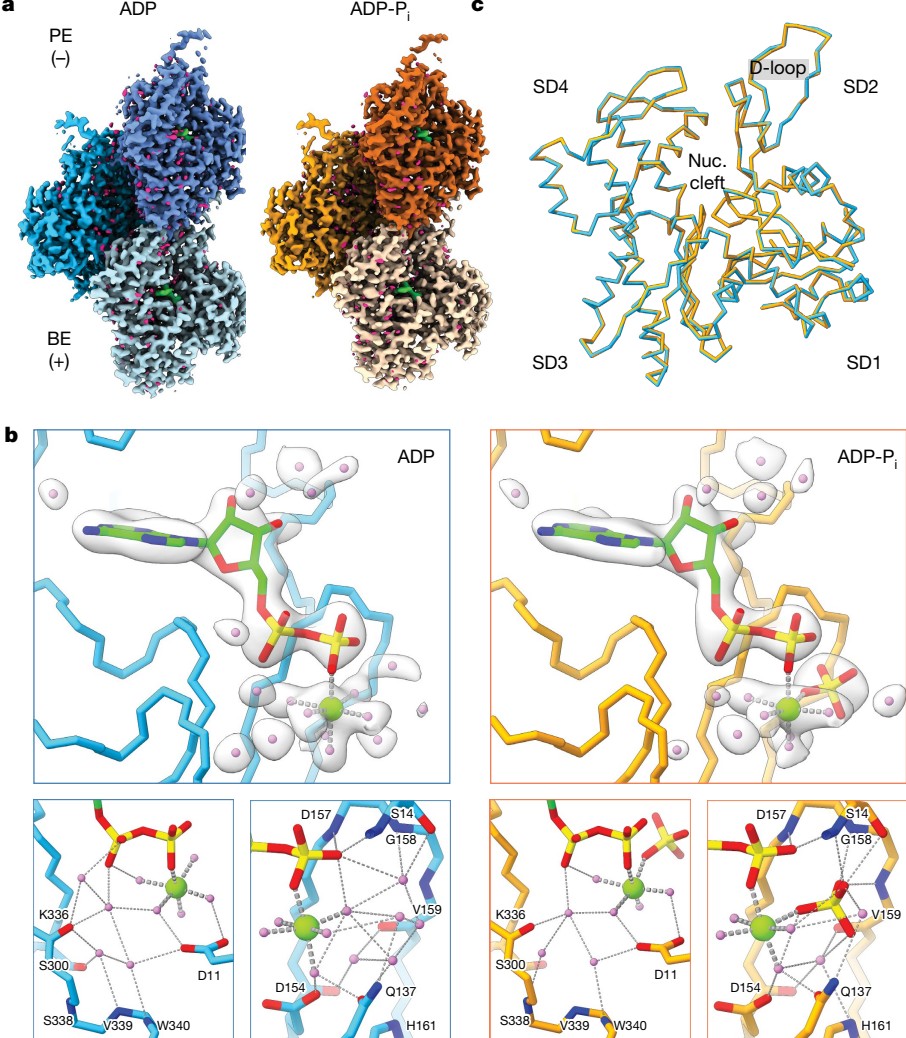

**Fig. 1 | Nucleotide cleft water networks are remodelled upon phosphate release by F-actin. a**, Cryo-EM maps of ADP–F-actin (left, shades of blue) and ADP-$P_i$–F-actin (right, shades of orange). ADP (green) and water (magenta) densities are shown. BE, barbed end; PE, pointed end. **b**, Atomic models of the ADP–F-actin (blue) and ADP-$P_i$–F-actin nucleotide (nuc.) clefts (orange) are shown in Cα representation. Top, carved transparent grey density is displayed for ADP (green), $PO_4^{3-}$ (yellow), $Mg^{2+}$ (light green) and water molecules (violet). In ADP/$PO_4^{3-}$, nitrogen atoms are blue, oxygen atoms are red and phosphorous atoms are yellow. Bottom, the backbone and side chain residues involved in putative hydrogen-bonding networks (dashed lines) are displayed and coloured by heteroatom. **c**, Superposition of individual ADP–F-actin (blue) and ADP-$P_i$–F-actin (orange) protomers, displayed in Cα representation.

(twist–bend coupling)[37]. Recent finer-grained simulations featuring subunits composed of multiple rigid bodies exhibited cooperativity between ATP hydrolysis and phosphate release through the filament lattice[38], suggesting coupling between propagation of lattice deformations and filament nucleotide state. However, in the absence of direct structural visualization, it remains unclear whether mechanically evoked conformational transitions in F-actin intersect with filament nucleotide state to control ABP engagement.

## Solvent in the nucleotide cleft of F-actin

To investigate whether subtle F-actin conformational changes could explain ABP nucleotide-state sensing, we determined the structures of ADP–F-actin and ADP-$P_i$–F-actin at an improved resolution. As reported protocols for preparing ADP-$P_i$–F-actin vary[13,27,32], we validated our approach (polymerizing G-actin in the presence of 15 mM $KH_2PO_4$) by monitoring cofilin severing using a fluorescence microscopy assay (Methods). Consistent with previous reports[14,15], ADP–F-actin rapidly depolymerized within 5 min of cofilin addition, whereas ADP-$P_i$–F-actin largely remained intact. Replacing

$KH_2PO_4$ with $K_2SO_4$ produced intermediate severing, suggesting that phosphate specifically caused the substantially reduced severing of ADP-$P_i$–F-actin rather than altering the buffer's ionic strength (Extended Data Fig. 1a,b). We therefore used cryo-EM and iterative helical real-space refinement as implemented in RELION to determine the structures of ADP–F-actin and ADP-$P_i$–F-actin at resolutions of 2.4 Å and 2.5 Å, respectively (Fig. 1a, Methods, Extended Data Fig. 1c,d and Supplementary Table 1).

These maps featured sufficient resolution to accurately build and refine ADP, magnesium, phosphate and water molecules (Fig. 1b, Methods, Extended Data Fig. 1e and Supplementary Video 1). In both nucleotide states, extensive water-mediated hydrogen-bonding networks stabilize the ligands. Notably, in ADP–F-actin, four water molecules displace the inorganic phosphate's oxygen tetrahedron. Other nucleotide cleft positions also featured small changes in the number and positioning of water molecules. These F-actin active-site compositions and stereochemistry are consistent with a contemporary study[39], providing confidence in interpreting the cryo-EM maps.

Despite substantial changes in bound small molecules, differences between the conformations of actin protomers themselves,

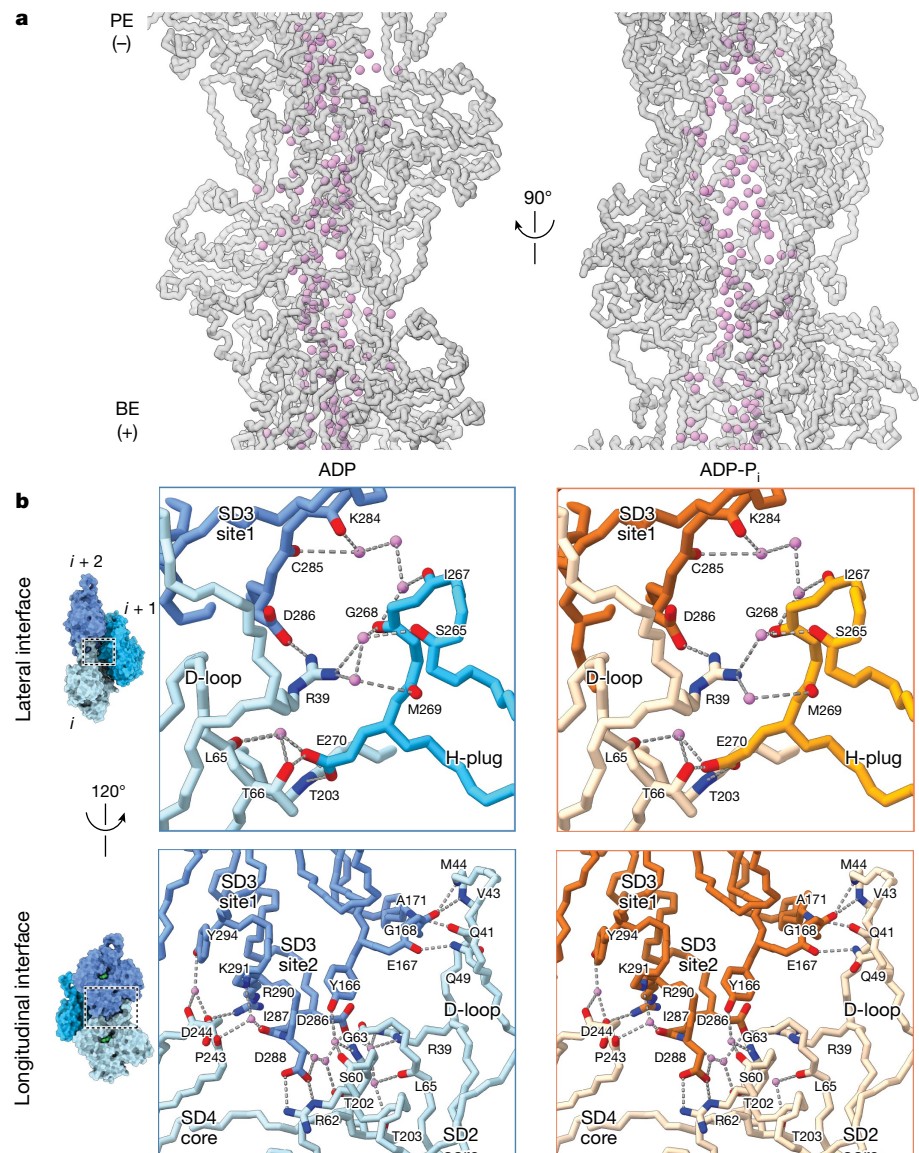

**Fig. 2 | Water molecules mediate key longitudinal and lateral contacts in F-actin. a**, Water molecules (violet) contained within the filament core of ADP–F-actin. Actin subunits are shown in transparent grey main-chain representation. **b**, Solvent-mediated contacts at lateral (top) and longitudinal (bottom) interfaces in ADP–F-actin (shades of blue) and ADP-P$_i$–F-actin (shades of orange). Key side chains and backbone atoms are displayed and coloured by heteroatom. Water molecules are shown in violet, and putative hydrogen bonds are indicated by dashed lines. Protomer indices are indicated.

which were exquisitely resolved (Extended Data Fig. 2a), were miniscule (0.206 Å Cα root mean squared deviation (r.m.s.d.); Fig. 1c). Per-residue r.m.s.d. and per-residue strain pseudoenergy analysis (a metric that highlights local deformations; Methods) confirmed the absence of meaningful Cα deviations with the exception of the D-loop in subdomain 2 (Extended Data Fig. 2b,c). This region, which is well established to be flexible in F-actin[27,32,40,41], also corresponded to the lowest-resolution region of our maps, and differences may therefore be attributable to resolution-dependent uncertainty in the atomic models. At the helical-lattice level, the refined rise was slightly larger in ADP–F-actin (28.1 Å versus 27.8 Å in ADP-P$_i$–F-actin), which accumulates into detectable differences in subunit positioning at longer length scales (Extended Data Fig. 2d). Nevertheless, these structures did not reveal changes of the magnitude associated with ABP binding (for example, cofilin[33,34]). This suggests that nucleotide-state sensitivity in ABPs is unlikely to be mediated primarily by their detection of F-actin conformational changes concomitant with phosphate release.

## Water molecules mediate intersubunit contacts

We next examined water molecules outside the nucleotide cleft, reasoning that solvent at intersubunit interfaces could potentially facilitate mechanical remodelling of the filament. As is typical for helical reconstructions, the highest-resolution map regions were the filament cores (Extended Data Fig. 1d), which revealed a continuous solvated channel aligned with the filament axis. Water positioning in this channel was largely similar between ADP–F-actin and ADP-P$_i$–F-actin (Fig. 2a and Extended Data Fig. 2e). An analysis of solvent-accessible pockets (Methods) indicates that the filament core is accessible to bulk solvent and features small channels that connect to the nucleotide clefts of the protomers. This positions the filament core to contribute to extensive solvent-mediated lateral interactions between strands, which occur at a vertex between three subunits (Fig. 2b (top)). We also observed ordered water molecules on the outer surface of F-actin that mediate longitudinal interactions between protomers along the same strand (Fig. 2b (bottom)). A detailed analysis of specific interactions[42,43] is

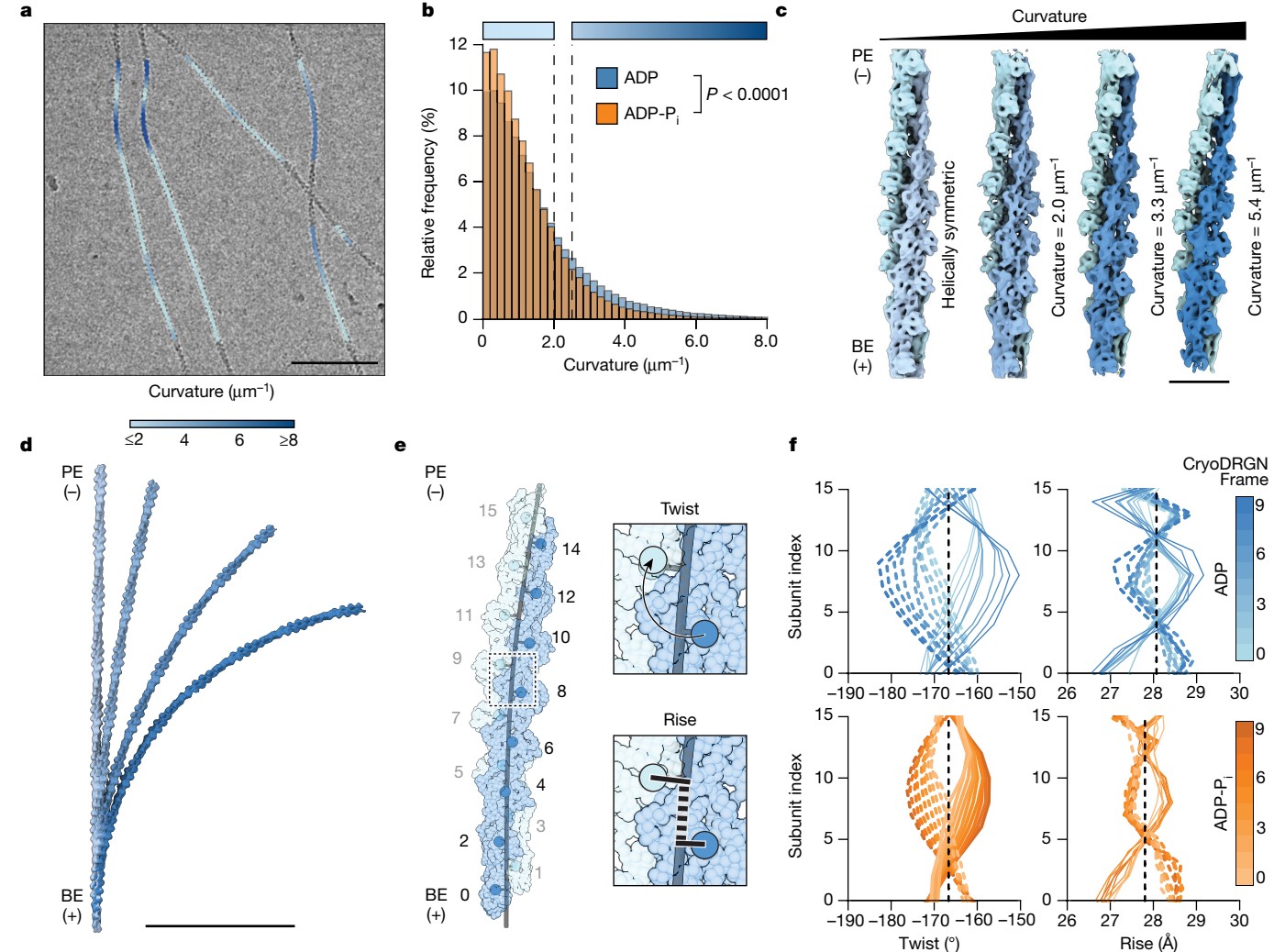

**Fig. 3 | Cryo-EM reconstructions of mechanically deformed F-actin reveal bend–twist coupling. a**, Representative cryo-EM micrograph of ADP–F-actin filaments featuring high-curvature regions, low-pass filtered to 30 Å. Picked segments are coloured by estimated curvature as indicated. Scale bar, 100 nm. **b**, Normalized curvature histograms of ADP–F-actin (blue, $n = 374{,}942$) and ADP-$P_i$–F-actin (orange, $n = 470{,}625$) filament segments, compared using a two-tailed Mann–Whitney $U$-test. The dashed lines indicate curvature thresholds for straight ($\leq 2.0\ \mu m^{-1}$) and bent ($\geq 2.5\ \mu m^{-1}$) segments. The colour bars correspond to the curvature key in **a**. **c**, Helically symmetric ADP–F-actin (left map) and cryoDRGN reconstructions sampling continuous bending of ADP–F-actin (right three maps), low-pass filtered to 8 Å. Strands are coloured in shades of blue. Scale bar, 10 nm. **d**, Stitched volumes of straight and bent maps from **c**, aligned to the bottom 16 protomers. Scale bar, 100 nm. **e**, Schematic of twist and rise measurements along a bent filament axis. Protomer numbering is indicated. **f**, Twist and rise measurements of cryoDRGN reconstructions sampled along the major variability component. The solid and dashed curves correspond to measurements from even-to-odd and odd-to-even protomer indices, respectively.

presented in the Supplementary Discussion. In summary, we found that intersubunit interfaces are extensively solvated, which we hypothesize could lubricate mechanical rearrangements within the filament.

## Visualizing F-actin bending deformations

Actin filaments are exposed to thermal fluctuations and fluid forces[12,44] during cryo-EM grid preparation, producing visibly bent regions (Fig. 3a) that are generally excluded during helical processing. Although other stably curved protein filaments have been structurally characterized[45,46], non-equilibrium bending has not been visualized in atomistic detail. As F-actin bending is energetically unfavourable, only a small filament segment subpopulation should feature appreciable deformations. Moreover, bending is a continuum, rendering it refractory to characterization by traditional classification approaches.

We therefore developed a neural-network-based approach to identify bent F-actin segments in micrographs (Methods and Extended Data

Fig. 3) and estimate their instantaneous in-plane curvature. As anticipated, most segments exhibited low curvature. However, a subset of filaments featured regions of continuous curvature gradients indicative of elastic bending (Fig. 3a). Consistent with persistence length measurements[35], segment curvature distributions revealed significantly greater mean curvature for ADP–F-actin compared with ADP-$P_i$–F-actin (ADP, $1.14\ \mu m^{-1}$; ADP-$P_i$, $0.96\ \mu m^{-1}$), including a long tail of highly curved ADP–F-actin segments (Fig. 3b). A minimal Boltzmann model of thermal fluctuations approximately captures the curvature distributions, yet systematic deviations are consistent with fluid forces during cryo-EM sample preparation increasing the prevalence of highly curved filaments[9,10,44] (Extended Data Fig. 4 and Supplementary Discussion).

To investigate these bending conformational landscapes, we examined 16-protomer segments featuring estimated curvatures greater than an arbitrary $2.5\ \mu m^{-1}$ cut-off (hereafter, bent F-actin). After ab initio reconstruction in cryoSPARC and subsequent processing with RELION, we used the heterogeneity analysis tool cryoDRGN to generate

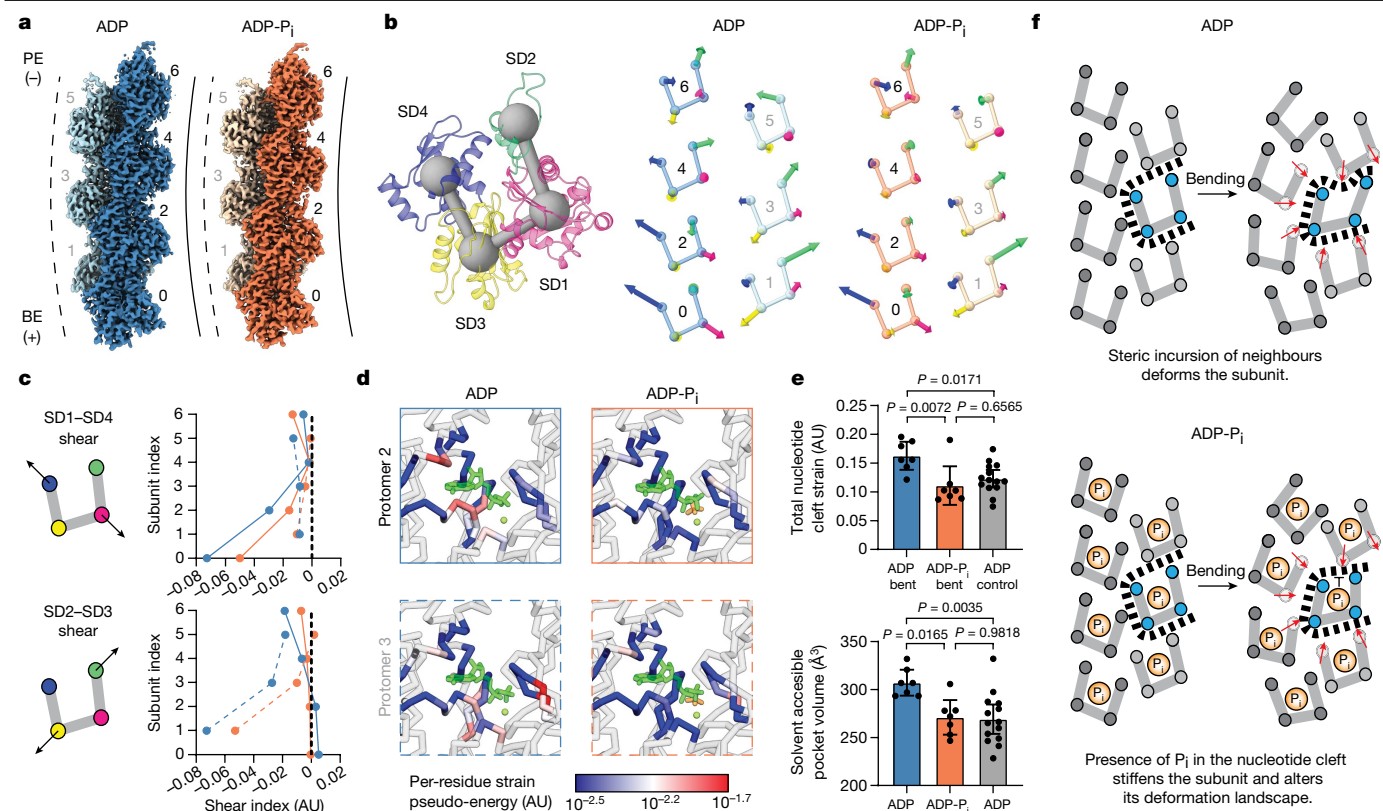

**Fig. 4 | Actin nucleotide state modulates subunit shearing during filament bending. a**, Cryo-EM maps of bent ADP–F-actin (blue) and ADP-P$_i$–F-actin (orange), coloured by strand. Protomer numbering is indicated. The dashed and solid lines represent the convex and concave sides of the bent filament, respectively. **b**, Ribbon representation of an individual actin protomer coloured by subdomain (left). The spheres connected by bars indicate subdomain centroids. Right, protomer subdomain centroid diagrams with vectors (scaled 100×), indicating subdomain-averaged displacements from the corresponding helically symmetric model. **c**, Plots of subdomain shear indices (representing coordinated rearrangements) of subdomains 1 and 4 (top) and subdomains 2 and 3 (bottom). Blue lines, ADP; orange lines, ADP-P$_i$. The solid and dashed lines represent even (concave side) and odd (convex side)

protomers, respectively. AU, arbitrary units. **d**, Cα representation of the indicated protomers' nucleotide clefts from bent ADP–F-actin and ADP-P$_i$–F-actin, coloured by per-residue strain pseudoenergy. ADP (dark green), magnesium (light green) and phosphate (orange) are shown in stick representation. **e**, Quantification of each protomer's nucleotide cleft strain pseudoenergy, compared between nucleotide states and bending conditions (top). Bottom, equivalent quantification of solvent-accessible volume of nucleotide clefts. Data are mean ± 95% confidence interval. $n = 7$ (bent) and $n = 14$ (straight). Statistical comparison was performed using one-way analysis of variance with Tukey post hoc analysis; $P$ values were corrected for multiple comparisons. **f**, Cartoon model of the steric boundary mechanism for joint regulation of F-actin by bending forces and nucleotide state.

volumes spanning each dataset's continuous bending deformations (Fig. 3c,d, Methods and Extended Data Fig. 5). CryoDRGN does not generate independent half-maps for resolution assessment, but the maps featured well-resolved α-helices characteristic of ~8 Å resolution. These reconstructions predominantly exhibited in-plane bending (Fig. 3d and Extended Data Fig. 5b,e), featuring central axis curvatures that fell within the distributions of estimated segment curvatures (ADP, 2.0–5.4 µm⁻¹; ADP-P$_i$, 3.2–4.4 µm⁻¹). Consistently, the radii of curvature of volumes stitched from multiple copies of these reconstructions were on the order of hundreds of nanometres (Fig. 3d), similar to bent F-actin observed in cells[7–10].

## Bending remodels F-actin's lattice

To examine lattice remodelling, we flexibly fit atomic models into these maps using ISOLDE and measured instantaneous helical parameters (rise and twist; Fig. 3e and Supplementary Video 2) along their bent central axes (Methods). This revealed striking bend–twist coupling, consistent with theoretical predictions[37], with alternating overtwisting and undertwisting of strands that increased with filament axis curvature. Symmetric twist deviations between strands remain in-phase, retaining canonical F-actin's twist as the instantaneous average between

odd/even protomers to maintain lattice structural integrity. Bending evoked twist deviations up to 15°, substantially greater than the twist deviation induced by cofilin (~5°)[33,34]. Each strand's bend–twist relationship could be modelled analytically as a travelling wave along its component protomers (Methods), whereby increasing curvature modulates the twist amplitude and phase (Fig. 3f, Extended Data Fig. 6a and Supplementary Table 2). Both ADP–F-actin and ADP-P$_i$–F-actin exhibit nearly identical propagation speeds in twist/protomer-index space, but ADP–F-actin features a larger-amplitude coupling factor. This leads to the physical interpretation that a given curvature produces greater twist deviations in ADP–F-actin than ADP-P$_i$–F-actin, suggesting ADP F-actin's lattice is more deformable (Extended Data Fig. 6b and Supplementary Table 2).

We also observed modest rise deviations that manifest as apparent standing rather than travelling waves, of which the amplitude also increases with curvature (Fig. 3f). These rise waves have wavelengths approximately half those of the twist waves and, to our knowledge, they have not been theoretically modelled in detail. Although the physical basis of rise deviations is less clear, we hypothesize that they result from subunit deformations to accommodate twist remodelling and increased elastic energy at intersubunit interfaces. Consistently, the most highly curved 16-protomer models featured nucleotide-state-dependent

alterations in the distances and angles between subdomains of longitudinally adjacent protomers along each strand[47] (Extended Data Fig. 6c; a detailed analysis is provided in the Supplementary Discussion). Notably, bent F-actin exhibited no clear systematic differences associated with subunit flattening, the hallmark of the G-to-F transition, in either nucleotide state, suggesting that F-actin bending has a distinct structural landscape.

To validate the cryoDRGN results, we divided the ADP segments into three straight bins, one low-curvature bin and one high-curvature bin, all of approximately equivalent size. Asymmetric single-particle reconstructions (Methods) of the straight controls (7.0 Å, 7.1 Å and 7.2 Å resolution) revealed negligible deviations from the helical parameters of canonical F-actin, whereas the low-curvature (6.9 Å resolution) and high-curvature (6.4 Å) reconstructions exhibited twist and rise patterns consistent with the cryoDRGN models (Extended Data Fig. 5d,f,g and Supplementary Video 3). Collectively, these data suggest that bending evokes substantial, nucleotide-state-modulated F-actin conformational transitions.

## Nucleotide clefts feature shear strain

We next sought to visualize protomer structural deformations accompanying filament bending. Reasoning that continuous conformational flexibility was limiting the resolution of the 16-protomer reconstruction, we focused on the central 7 subunits (Methods). Focused refinement substantially improved the resolution of both bent F-actin reconstructions (ADP–F-actin, 3.6 Å; ADP-P$_i$–F-actin, 3.7 Å; Fig. 4a, Extended Data Fig. 7a–c and Supplementary Table 3), facilitating direct atomic model building and refinement. We also reconstructed two straight ADP–F-actin bins featuring matched segment numbers—both yielded 3.7 Å maps, into which we built control atomic models (Extended Data Fig. 7 and Supplementary Table 3).

We compared these asymmetric models with our helically symmetric models by superimposing their central protomers, revealing systematic rearrangements solely in the bent models (Extended Data Fig. 7d). The straight controls featured only small, randomly distributed deviations, suggesting that the bent structures capture conformational rearrangements rather than merely reflecting model building uncertainty at 3.6–3.7 Å resolution. To examine the internal deformations of individual subunits, we superimposed each protomer with a subunit from its nucleotide-state-matched helically symmetric model and examined subdomain-averaged Cα displacements (Fig. 4b, Methods and Supplementary Video 4). This revealed complex displacement patterns that were dependent on a protomer's strand membership and lattice position, primarily characterized by subunit shearing around the nucleotide cleft. These rearrangements are captured by a shear index (Fig. 4c; the dot product of the indicated vectors), which reveals increased shearing near the barbed ends of our reconstructions, probably due to the local orientation of the lattices relative to the plane of curvature. Although subdomain displacement patterns were highly similar between ADP and ADP-P$_i$ (Fig. 4b), the shearing magnitude was greater in ADP (Fig. 4c), consistent with the higher overall average curvature of bent ADP segments and enhanced deformability of ADP subunits.

To identify specific structural elements experiencing deformations, we once again performed per-residue strain pseudoenergy analysis (Methods), which is insensitive to rigid-body displacements. This revealed three primary sites of strain: the H-plug (Extended Data Fig. 8a (left)), the D-loop (Extended Data Fig. 8a (right)) and the nucleotide cleft (Fig. 4d). Strain in the H-plug and D-loop is consistent with their roles as primary mediators of intersubunit interactions. Notably, bent ADP–F-actin displayed significantly higher total strain in residues adjacent to the nucleotide (7.5 Å distance cut-off) compared with bent ADP-P$_i$–F-actin or the straight ADP–F-actin controls (Fig. 4d,e). Consistently, solvent-accessible volume measurements (Methods) also revealed expanded nucleotide clefts in the bent ADP–F-actin protomers relative to bent ADP-P$_i$–F-actin and straight ADP–F-actin controls (Fig. 4e). We

also examined detailed rearrangements within the subunits (Extended Data Figs. 8b and 9, Supplementary Discussion and Supplementary Videos 5 and 6). These were broadly consistent with actin's nucleotide cleft functioning as a deformable locus that coordinates mechanical rearrangements, the rigidity of which is dependent on phosphate occupancy. This provides a structural mechanistic explanation for actin nucleotide state's modulation of F-actin bending mechanics.

## Discussion

Our direct structural visualization of a mechanically regulated F-actin conformational landscape modulated by actin nucleotide state reveals substantial remodelling of helical lattice twist, producing substantial rearrangements at protomer–protomer contacts. We therefore speculate that F-actin bending could modulate binding by numerous ABPs, as their binding sites generally span two longitudinally adjacent protomers. Indeed, mapping the known binding sites of representative force- and nucleotide-state sensitive ABPs on bent F-actin suggests that they are likely to be impacted by these structural transitions (Extended Data Fig. 10). Although it is possible that nucleotide-state-sensing ABPs detect the very small structural changes (most plausibly, the 0.3 Å rise increase) allosterically triggered by phosphate release from ADP-P$_i$–F-actin[22,27], our high-resolution structures suggest that this is unlikely to be the primary mechanism. Our bent actin structures support an alternative model in which phosphate both rigidifies F-actin (consistent with previous studies)[33] and modifies the structural landscape evoked by bending forces, enabling discrimination by ABPs. This model is furthermore compatible with the rigidification of canonical F-actin by phosphate inhibiting engagement by ABPs that must substantially deform the lattice to bind, such as cofilin[15,32,48].

Our studies also provide insights into the mechanisms of F-actin bending, a model for regulation of protein structure by mechanical force. As force is a delocalized perturbation across the filament, it has been unclear how it is transduced into conformational remodelling of component subunits. We propose a 'steric boundaries' conceptual model of mechanical regulation, wherein lattice rearrangements reposition a subunit's contacting neighbours (Fig. 4f). This remodels the physical space available for it to occupy, inducing it to deform to minimize clashes. The model predicts coupling between a subunit's lattice position, local filament curvature and its resultant conformation, consistent with our observations. Ligand binding/chemical modifications can also alter a subunit's deformation landscape, providing a framework for intersecting biochemical and mechanical regulation. As shown here for actin and phosphate, this does not require ligand engagement to modify the protein's ground-state conformation, a departure from traditional allosteric regulation that bears resemblance to dynamic allostery[49]. However, in the steric boundaries framework, the ligand co-regulates mechanical deformations rather than altering the protein's intrinsic conformational fluctuations (a non-exclusive mode of regulation). Consistently, we found that actin's nucleotide cleft mediates subunit shearing deformations, enabling its biochemical content to modulate F-actin's mechanics. This additionally suggests that force could modulate F-actin's nucleotide hydrolysis and phosphate release kinetics, as has previously been predicted[50]. Beyond F-actin, we anticipate the steric boundaries mechanism could also explain joint mechanical and biochemical regulation of other multisubunit complexes—a subject for future research.

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

## Methods

### Protein preparation

Chicken skeletal muscle actin was purified as previously described[51]. In brief, 1 g of chicken skeletal muscle acetone powder was resuspended in 20 ml of G-Ca buffer (G buffer: 2 mM Tris-Cl pH 8.0, 0.5 mM DTT, 0.2 M ATP, 0.01% NaN$_3$, supplemented with 0.1 mM CaCl$_2$) and mixed by inversion for 30 min. The suspension was centrifuged in a Beckman Ti70 rotor at 42,500 rpm (79,766$g$) for 30 min. Then, 50 mM KCl and 2 mM MgCl$_2$ were added to the supernatant, containing G-actin monomers, to stimulate F-actin polymerization for 1 h. KCl (0.8 M) was then added, and the solution was incubated for 30 min to facilitate dissociation of contaminants from F-actin. The solution was then centrifuged in a Ti70 rotor at 42,500 rpm (79,766$g$) for 3 h. The pellet was resuspended in 2 ml of G-Ca buffer and incubated overnight. The mixture was then homogenized in a Dounce chamber for 10–15 passes, consecutively sheared through 26G and 30G needles, then dialysed in 1 l of G-Ca buffer overnight in Spectra/Por 1 dialysis tubing (MWCO 6–8 kDa). The actin solution was then sheared through a 30G needle again before dialysis in 1 l of fresh G-Ca buffer for another day. It was then centrifuged in a Beckman Ti90 rotor at 70,000 rpm (187,354$g$) for 3 h. The top two-thirds of the supernatant was then loaded onto a HiLoad 16/600 Superdex 200 column (Cytiva) for size-exclusion chromatography. Purified G-actin was maintained in G-Ca buffer at 4 °C before use.

A Flag–GFP tagged version of human myosin VI-S1 (used to anchor actin filaments to coverslips for cofilin severing assays) was purified as described previously[24], flash-frozen in liquid nitrogen and maintained at −80 °C. Lyophilized human cofilin 1 was purchased from Cytoskeleton (CF01) and reconstituted in MB buffer (20 mM MOPS pH 7.4, 5 mM MgCl$_2$, 0.1 mM EGTA, 50 mM KCl, 1 mM DTT), then incubated overnight at 4 °C. Aliquots were then flash-frozen in liquid nitrogen and maintained at −80 °C. Aliquots (20 µg) of lyophilized rhodamine actin (Cytoskeleton AR05) were resuspended in 18 µl of G-Ca buffer and 2 µl of MilliQ water, incubated at 4 °C for at least 1 h, then clarified by ultracentrifugation in a Beckman TLA100 rotor at 100,000 rpm (335,400$g$) for 20 min.

### Cofilin severing assays

Glass coverslips (Corning 22 × 50 mm, 1½) were cleaned for 30 min using 100% acetone, 10 min using 100% ethanol and 2 h using 2% Hellmanex III liquid cleaning concentrate (HellmaAnalytics) in a bath sonicator followed by rinsing with MilliQ water. The cleaned glass coverslips were coated with 1 mg ml$^{-1}$ mPEG5K-Silane (Sigma-Aldrich) in a 96% ethanol, 10 mM HCl solution for at least 16 h with rocking. After coating, the coverslips were rinsed with 96% ethanol and water, then air-dried and stored at 4 °C until use.

Rhodamine-labelled ADP–F-actin (20%) was prepared by diluting unlabelled G-actin and rhodamine G-actin stocks to 0.9 µM and 0.1 µM, respectively, in KMEH buffer (50 mM KCl, 1 mM MgCl$_2$, 1 mM EGTA, 10 mM HEPES pH 7.0, 1 mM DTT) supplemented with G-Mg (G Buffer + 0.1 mM MgCl$_2$), for a final actin concentration of 1 µM. The mixture was incubated at 25 °C for 1 h, then 4 °C overnight. Before use, the F-actin was pelleted by centrifugation at 60,000 rpm (120,744$g$) in a TLA100 rotor for 20 min and resuspended in fresh KMEH to remove any free phosphate ions. Rhodamine-labelled ADP-P$_i$–F-actin (20%) was prepared as described above, except KMEH + G-Mg was supplemented with 15 mM K$_2$HPO$_4$ (pH 7.0). The mixture was incubated at 25 °C for 1 h, then placed on ice and used immediately. Rhodamine-labelled ADP–F-actin (20%) in the presence of K$_2$SO$_4$ was prepared identically, except 15 mM K$_2$SO$_4$ (pH 7.0) was substituted for K$_2$HPO$_4$.

For preparation of total internal reflection fluorescence (TIRF) samples, a PDMS gasket (Grace Bio-Labs, 103380) was placed onto the cover slip and 20 µl of 0.25 µM rigor myosin VI S1 in MB buffer was added to the well and incubated for 2 min, followed by blocking with 20 µl of 0.1% polyvinylpyrrolidone (Sigma-Aldrich, 9003-39-8) in MB buffer

for 1 min. Next, 20 µl of 1 µM F-actin was added to the well for 30 s. The actin was then washed with 20 µl MB buffer (for ADP–F-actin) or MB buffer + 15 mM KH$_2$PO$_4$/K$_2$SO$_4$ (for ADP-P$_i$–F-actin/ADP–F-actin in the presence of sulfate).

TIRF videos were recorded using Nikon's NIS-Elements software at either a 1 s or 2 s frame rate on a Nikon H-TIRF system using a CFI Apo ×60 TIRF oil-immersion objective (NA 1.49), a quad filter (Chroma) and an iXon EMCCD camera (Andor). Rhodamine was excited by a 561 nm laser. Filaments were initially imaged for 2 min, the video was paused, and 20 µl of 2 µM cofilin in MB buffer (1 µM final concentration) was added to the well. The video was then resumed and filament severing was recorded for an additional 8 min.

### Cofilin severing quantification

Videos were analysed using custom Python scripts that measured the change in filament intensity over the course of the experiments. Video regions containing F-actin were identified and masks were generated by operating on a projection of the first 50 frames of the video using the functions in the scikit-image Python package[52]. This projection's background (computed using a rolling ball radius of 50 pixels) was subtracted and subjected to a gaussian blur with a filter size of 2 pixels. A Li adaptive threshold was used to binarize the projection, morphological objects with an area smaller than 100 pixels were removed and the remaining binarized image was dilated by 1 pixel. This set of masks for each video was applied to all frames of the video, and quantification of actin intensity was performed on a per-mask basis.

For each mask, the summed pixel intensity was measured for each frame and normalized by dividing by the 90th percentile intensity. The maximum intensity was not used for normalization because the intensity often spiked with the addition of buffer/cofilin at time 0 s. The intensity traces for each mask of each video of the same experimental condition were pooled and their average was plotted (Extended Data Fig. 1a,b).

### Cryo-EM grid preparation

ADP-P$_i$–F-actin was prepared as described above (without incorporation of rhodamine actin), then diluted to 0.5 µM in KMEH + 15 mM KH$_2$PO$_4$ supplemented with 0.01% Nonidet P40 (NP40) substitute (Roche), an additive that we have found improves our ability to achieve thin vitreous ice films. Solution (3 µl) was applied to a plasma-cleaned C-flat 1.2/1.3 holey carbon Au 300 mesh grid (Electron Microscopy Sciences) in a Leica EM GP plunge freezer operating at 25 °C. After incubation for 60 s, the grid was blotted from the back using a Whatman no. 5 filter paper for 4 s, then flash-frozen in liquid ethane.

The ADP–F-actin sample corresponds to a pre-existing dataset described in a recent study[41]. ADP–F-actin was prepared as described above, except KH$_2$PO$_4$ was omitted and KMEI buffer (50 mM KCl, 1 mM MgCl$_2$, 1 mM EGTA, 10 mM imidazole pH 7.0, 1 mM DTT) + 0.01% NP40 substitute was used instead of KMEH.

### Cryo-EM data collection

ADP–F-actin and ADP-P$_i$–F-actin datasets were collected on the same FEI Titan Krios system operating at 300 kV and equipped with a Gatan K2-Summit direct electron detector using super-resolution mode. Videos were recorded using the SerialEM software suite[53] at a nominal magnification of ×29,000, corresponding to a calibrated pixel size of 1.03 Å at the specimen level (super-resolution pixel size of 0.515 Å per pixel). Each 10 s exposure was dose-fractionated across 40 frames, with a total electron dose of 60 e$^-$ Å$^{-2}$ (1.5 e$^-$ Å$^{-2}$ per frame), with defocus values ranging from −1.5 to −3.5 µm underfocus. For the ADP-P$_i$–F-actin dataset, beam-image shift was used to collect 4,834 single exposures from nine holes in a 3-by-3 grid per each stage translation. For the ADP–F-actin dataset, which has previously been reported[41] and was reprocessed here, 1,548 exposures were directly targeted using stage translations with a single exposure per hole.

## Micrograph pre-processing

Movies were aligned with MotionCor2 using 5 × 5 patches[54], and dose-weighting sums[55] were generated from twofold binned frames with Fourier cropping, resulting in a pixel size of 1.03 Å in the images. Non-dose-weighted sums were used for contrast transfer function (CTF) parameter estimation using CTFFIND4[56].

## Synthetic dataset generation

Accurate nanoscale curvature measurements of F-actin in noisy cryo-EM micrographs requires high-quality, pixel-wise image segmentation. Traditional cross-correlation-based approaches for filament particle picking use templates derived from 2D class averages or projections of a straight F-actin map. This strategy features limitations, notably that cross-correlation will be lower between straight templates and highly curved filament segments in experimental images. Moreover, discrimination of filaments from background or non-protein signal may be poor. To achieve high-quality semantic segmentation, we implemented a convolutional neural-network-based approach to identify filament segments of all curvatures. Although other machine-learning-based pickers have recently been introduced[57–59], to our knowledge, they do not explicitly focus on detecting or flagging instantaneous curvature within a filamentous assembly. From semantically segmented micrographs, we identified filaments and measured their instantaneous in-plane 2D curvature.

To train the neural networks, synthetic pairs of noisy and noiseless projections that mimicked experimental data were used. Plausible 3D synthetic models of F-actin bent around a circular central axis were generated using a custom Python script that loaded and operated on Protein Data Bank (PDB) models using functions from the ProDy package[60]. Individual actin protomers were treated as rigid objects and positioned using a toroidal helix function:

$$\gamma(t) = \begin{pmatrix} r\cos(\omega t + \varphi) + d_1 \\ (\cos(c\,t/R))(r\sin(\omega t + \varphi) + d_2 + R) - R \\ (\sin(c\,t/R))(r\sin(\omega t + \varphi) + d_2 + R) + d_3 \end{pmatrix},$$

where the parameters are defined as follows: $\gamma$ is the position in 3D space along the toroid, $r$ is the filament radius, $\omega$ is the average twist, $t$ is the parameterized position along the helical curve, $\varphi$ is the phase of the twist, $d_1$, $d_2$, and $d_3$ are the displacements of the toroid from the origin, $c$ is the rise parameter and $R$ is the toroid's radius of curvature. Note that this function converges to a canonical F-actin helix when the curvature is zero. Furthermore, the equation does not explicitly encode emergent architectural remodelling phenomena such as twist–bend coupling. Using this synthetic filament generation scheme, a library of 135 bent actin models consisting of 35 protomers and of systematically varying curvature and rotation about the central filament axis were generated. These models were then converted to volume files using the PDB2MRC function in EMAN2[61]. These volumes were saved within 256-voxel boxes (voxel size, 4.12 Å). The volumes were rotated about the phi and rot angles by a random, uniformly sampled value between 0° and 359°, and the tilt was randomly sampled from a Gaussian probability distribution centred at 90° with a s.d. of 2.5°, then randomly translated around the box by ±250 Å and projected along the z axis to generate a noiseless projection. A paired noisy projection was generated by adding pink noise in Fourier space, as implemented in EMAN2's Python package to generate realistic-looking synthetic data[61]. These projection images were cropped to a smaller box size of 128 pixels to ensure that filaments would span the image. Two-channel stacks of semantic maps associated with the noisy–noiseless projection pairs were generated by low-pass filtering the noiseless projection to 40 Å and binarizing it.

## Network architecture and training

A denoising autoencoder (DAE) was trained using the architecture outlined in Extended Data Fig. 3a. Each trainable layer had a ReLU activation function, except for the final layer, which had a linear activation function. The negative of the cross-correlation coefficient was used as the loss function. For training, the weights were initialized using the default initialization in TensorFlow[62]. The model was trained using the Adam optimizer version of stochastic gradient descent with a learning rate of 0.00005 and minibatch size of 16 until the model converged (no improvement in validation loss for 3 epochs). After network convergence, the weights from the best epoch were restored. For training, 800,000 noisy–noiseless projection pairs with box sizes of 128 × 128 were generated, 90% of which were used for training and 10% for validation. After network convergence, the DAE had an average cross-correlation coefficient of 0.9887 on the validation set.

After training the model as a DAE, a semantic segmentation network was trained by copying the convolutional encoding layers and weights of the DAE while adding convolutional layers. The final layer was a two-channel layer with sigmoid activation and default Tensor-Flow initialization. This semantic segmentation network was then trained with a learning rate of 0.001. For training, 30,000 pairs of noisy inputs and semantically segmented targets of dimension 128 × 128 and 128 × 128 × 2, respectively, were used with a minibatch size of 32; 90% of the synthetic data were used for training and 10% for validation. The loss function was binary cross-entropy and, after network convergence, the model had a loss of 0.0651 on the validation set. Example network performance on synthetic data is shown in Extended Data Fig. 3.

Models were trained on a single NVIDIA Titan XP GPU with 12 GB of VRAM. Training required approximately 1 h per epoch for the DAE and 3 min per epoch for the semantic segmentation network. After initiating this project, we continued developing deep-learning-based filament particle pickers. The architectures described here have been superseded by a U-net architecture, which we found produces better segmentation with a smaller training set in a shorter time[63].

## Particle picking

A custom Python script was used to pass images to the fully convolutional neural network for semantic segmentation (FCN-SS) and execute curvature-sensitive filament picking. Each micrograph was binned by 4 to a pixel size of 4.12 Å per pixel, then 128-pixel tiles featuring 32 pixels of overlap were extracted and passed as inputs to the network. The outputs were stitched together by maximum intensity projection at the overlaps, producing a semantic segmentation map of the micrograph. These maps were then binarized using a fixed, empirically determined threshold of 0.9 and skeletonized. Branches shorter than 8 pixels were pruned, and pixels within a radius of 48 pixels from filament intersections were removed. Continuous filaments were then identified by matching tracks with common end points, and 2D splines were fit through the filaments for curvature estimation. To prevent spuriously high curvature values due to edge effects, the terminal 50 pixels of the spline were omitted from picking. From the remaining filament sections, the instantaneous curvature was measured along the spline at 56 Å intervals (corresponding to a step size of the length of one protomer) and used for segment selection. For picking segments from the identified filaments for asymmetric reconstructions, a step size equivalent to two short-pitch helical rise steps (56 Å) was used for extracting segments. Filament segments with a curvature greater than or equal to 2.5 μm$^{-1}$ were considered to be bent, whereas those with a curvature of less than or equal to 2.0 μm$^{-1}$ were considered to be straight. To select segments from the identified filaments for high-resolution helical reconstructions, a step size of three times the helical rise was used (83.4 Å), and segments that were members of the same filament were flagged in the output metadata (a RELION-formatted STAR file).

## Helical image processing

High-resolution reconstructions were determined using the iterative helical real space refinement[64] approach as implemented in RELION[65]. We reprocessed our ADP–F-actin dataset that previously produced a map at 2.8 Å resolution[41], and the ADP-$P_i$–F-actin dataset from this work in parallel. Our neural-network-based picker eliminated intersections where filaments overlapped. After initial picking, particles were extracted without binning in 512-pixel boxes using RELION, with 81 Å (3 protomers) of non-overlap. Our picking scheme did not include psi (in-plane rotation) angle estimates, so an initial refinement was performed with global angular searches and a bare actin reference (EMD-24321) low-pass filtered to 35 Å. After this initial alignment, the psi angles were changed to a psi prior, all poses were removed from the metadata file and the tilt prior was set to 90. This dataset was then processed for a previously described F-actin cryo-EM data processing workflow[23,41] in RELION-3.1[66], with minor modifications described below. In brief, initial 2D classification was performed to remove junk particles (only 0.4% of picked particles for ADP–F-actin and 11.2% of particles for ADP-$P_i$–F-actin), followed by 3D classification with alignment and five classes. For alignment, a search range of 15° around the tilt and psi priors was used, and global searches of the rot angle with an angular sampling of 7.5° was used. For ADP–F-actin, no particles were excluded at the 3D classification stage because all five classes were of high quality. For ADP-$P_i$–F-actin, two classes comprising 23% of the remaining segments (128,533 segments total) were excluded because their helical parameters were at the border of the search range and the classes appeared to be abnormal. Selected particles were then processed for unmasked 3D auto-refinement using the same angular search range described above (with local angular sampling starting at 1.7°) with helical symmetry searches as implemented in RELION-3.1. This yielded a map at 4.2 Å-resolution for ADP–F-actin and 4.1 Å-resolution map for ADP-$P_i$–F-actin. Post-processing was performed using a loose mask trimmed to 50% of the box size along the helical axis (z-length), which resulted in a resolution of 3.4 Å for ADP–F-actin and 3.5 Å for ADP-$P_i$–F-actin.

Then, several iterative rounds of CTF refinement, Bayesian polishing and 3D auto-refinement were performed. For both datasets, CTF refinement was initially performed by estimating the anisotropic magnification for each optics group. Defocus values were next fit on a per-particle basis and astigmatism was fit on a per-micrograph basis, along with beam tilt estimation. For the ADP-$P_i$–F-actin dataset, because beam-image shift was used during data collection, the data were processed in nine optics groups. Only a single optics group was used for the ADP–F-actin dataset, which was collected with stage translations. After CTF refinement, Bayesian polishing was used to improve the data's motion correction on a per-particle basis. The initial helical parameters for RELION's symmetry search were updated and the mask used for post-processing was used to run another round of 3D auto-refinement. This process was repeated using a 30% z-length mask, including estimation of trefoil and fourth order aberrations during CTF refinement. After the second round of particle polishing, a third round of CTF refinement was performed.

After the last iteration of CTF refinement, estimated defocus values were smoothed over the length of each continuous filament, similar to previously reported approaches[46]. Finally, a single round of masked refinement using a 30% z-length mask with local angular and translational searches was performed using solvent-flattened Fourier shell correlation (FSC) resolution assessment. The final reconstructions converged with resolutions of 2.43 Å for ADP–F-actin and 2.51 Å for ADP-$P_i$–F-actin.

## Image processing of bent F-actin

Selected bent segments were extracted in RELION with a box size of 512 × 512 pixels and pixel size of 1.03 Å per pixel (bin 1), initially with filament overlap. To avoid reference bias, ab initio initial model generation was performed using cryoSPARC[67] (v.2.11.0) using the subset of ADP–F-actin segments with an estimated curvature of greater than 4.0 μm$^{-1}$. Subsequent homogeneous refinement of these particles in cryoSPARC produced an asymmetric map with clear curvature. The data were then imported into RELION-3.0, and 2D classification without alignment was performed to remove junk particles, followed by unsupervised 3D classification with three classes using global angular searches. Two clearly bent, low-resolution classes curved in opposite directions and one junk class were produced. The particles in the bent classes were then processed for supervised classification using the two bent classes as references and one straight F-actin reference as a decoy using global angular alignment. Only 0.3% of particles were assigned to the decoy, consistent with the selected segments almost exclusively featuring bent F-actin. Alignment of the two bent classes revealed that they were nearly identical but displaced by one protomer, making them appear to bend in opposite directions. Their particles were therefore pooled for homogeneous refinement in cryoSPARC using global searches and the first bent class as a reference, low-pass filtered to 30 Å. These particles were then reimported to RELION, and underwent 3D auto-refinement using the cryoSPARC map low-pass filtered to 10 Å as a reference, local angular searches, a loose 70% z-length mask and solvent-flattened FSCs. This process was repeated to generate a less-bent map from segments with measured curvatures in the 2.5 μm$^{-1}$ to 4.0 μm$^{-1}$ range.

After demonstrating the feasibility of reconstructing bent filaments, all segments with a curvature above 2.5 μm$^{-1}$ were then processed for homogeneous refinement in cryoSPARC using the highly bent RELION refinement result as an initial reference, low-pass filtered to 30 Å. This was done separately for ADP–F-actin and ADP-$P_i$–F-actin in parallel. The data were then reimported into RELION for masked 3D auto-refinement using local searches. Successive rounds of CTF refinement, Bayesian particle polishing and 3D auto-refinement using a 70% z-length mask were performed until resolution improvement plateaued. For the bent ADP–F-actin dataset, four rounds of CTF refinement and three rounds of Bayesian polishing were performed. For the bent ADP-$P_i$–F-actin dataset, three rounds of CTF refinement and two rounds of Bayesian polishing were performed. From this stage, the data were either processed for high-resolution asymmetric analysis or continuous conformational variability analysis.

For high-resolution analysis, segment overlap within 360 pixels (corresponding to 7 protomers) was removed and the particles were processed for a final masked (70% z-length) 3D auto-refinement with local searches and solvent-flattened FSC calculations. Resolution anisotropy of these maps was assessed with the 3DFSC server[68]. For continuous conformational variability analysis, segment overlap within the entire 512-pixel box (corresponding to 16 protomers) was removed before final masked (90% z-length) 3D auto-refinement (also with local searches and solvent-flattened FSC calculations). These segments and their assigned poses were then used for training of neural networks in cryoDRGN[69] to assess conformational variability.

Asymmetric ADP–F-actin straight controls were generated using a similar method. All filament segments with a measured curvature less than or equal to 2.0 μm$^{-1}$ were subjected to ab initio map generation and homogeneous refinement in cryoSPARC. They were then imported to RELION for subsequent rounds of local 3D auto-refinement, CTF refinement and Bayesian polishing as described above. Then, all segment overlap was removed within 360 pixels, and two random subsets of sufficient size to generate maps of comparable resolution to the bent asymmetric maps were generated. These two subsets of particles underwent a final local 3D auto-refinement as described for the bent asymmetric reconstructions, and the resulting ~3.7 Å maps were used as controls for model building and analysis.

## Variability analysis of bent F-actin

To perform variability analysis on the consensus, 16-protomer asymmetric bent reconstructions, the particles were downsampled by 2 to a box size of 256 and a pixel size of 2.06 Å. Two cryoDRGN neural

networks were trained, one for the bent ADP–F-actin dataset and one for the bent ADP-P$_i$–F-actin dataset. In both cases, the network had a variational auto-encoder architecture of seven 1,024-dimensional encoding layers and seven 1,024-dimensional decoding layers with a 10-dimensional latent space. All of the other parameters were set to the default. The networks were trained for 40 epochs. Using four NVIDIA Titan XP GPUs, the average epoch time was ~12 min. Principal component analysis of the individual particle embeddings in the trained 10-dimensional latent space revealed the major variability present in the dataset to be flexible conformational heterogeneity due to bending deformations. Predicted reconstructions sampled along this trajectory in the latent space were used in subsequent analysis.

## High-resolution atomic models

Before model building and refinement, maps were processed for density modification with phenix.resolve_cryo_em[70], using only the maps as inputs, then resampled onto a grid featuring 0.2575 Å voxels through fourfold Fourier unbinning with the program resample.exe (distributed with FREALIGN[71]).

Our previous model of bare ADP–F-actin (PDB: 7R8V) was copied and rigid-body fit into the central three protomers in the ADP–F-actin and ADP-P$_i$–F-actin maps using UCSF Chimera[72]. Atomic models for each nucleotide state were then built and refined independently. Manual adjustments were made to the central protomer using Coot[73], and the other two chains were replaced with this updated protomer. These models, containing three actin protomers with associated ADP, and Mg$^{2+}$ (and PO$_4^{3-}$) ligands were refined using PHENIX real-space refinement[74] with non-crystallographic symmetry (NCS) restraints. After real-space refinement in phenix, initial solvent water molecules were placed using phenix.douse[75] with the mean scale parameter set to 0.4. Approximately 140 water molecules per protomer were initially placed with this automatic function. The maps and models were then manually inspected in Coot, and water molecules were added or pruned. As the map resolution decreased radially from the filament core, phenix.douse was unable to reliably detect water peaks in all of the map regions using a single threshold. After manual adjustments, all water molecules outside a central slab 28 Å along the filament axis (the approximate span of a single helical rise) were deleted. The water molecules within the slab were then symmetrized to make an 84-Å-long slab containing waters, and two protomer chains were added to the model to fully satisfy all neighbour contacts for the central protomer. Water molecules were then associated with the closest protomer. This central protomer was copied twice and each protomer with its ligands were then fit as rigid bodies into the map to form a new trimer model. Each water molecule was then manually inspected in Coot and adjusted to fit into the map density if needed. A final PHENIX real-space refinement was performed with NCS restraints on the protein chains but not the solvent waters. A summary of validation statistics is provided in Supplementary Table 1.

## Atomistic models from variability analysis

From each cryoDRGN frame, 16 copies of the corresponding helically symmetric central actin protomer model were rigid-body fit into the central 16 protomer sites in the map and combined into a single model. The map and initial model were then adjusted using the molecular dynamics flexible fitting[76]-based modelling software ISOLDE[77] implemented in ChimeraX[78], using secondary structure distance and torsional restraints. Owing to the relatively low resolution of the cryoDRGN predictions (by visual inspection estimated to be ~8 Å), the map weight was reduced to 10% of the automatically determined weight. The simulation temperature was set to 120 K, and the flexible fitting simulation was run for five real-time clock minutes before lowering the simulation temperature to 0 K and stopping it. These models were subsequently used for measurement of helical parameters and subdomain distance/angle measurements.

## Bent and control F-actin atomic models

Before model building, maps were processed for standard post-processing using RELION. Models were built into the ~3.6 Å asymmetric reconstructions by rigid-body fitting the central protomer from the corresponding helically symmetric model into each of the seven central protomer sites in the map along the filament length. Initial, large-scale adjustments to the model were performed using ISOLDE. For each condition, the map and model were loaded without applying any restraints, and the simulation temperature was set to 120 K. The simulation was then run for 5 min before lowering the simulation temperature to 0 K and ending the simulation. These models were then processed for PHENIX real-space refinement without using NCS. A summary of validation statistics is provided in Supplementary Table 3.

## Extended F-actin model for rise analysis

The extended 31-protomer helically symmetric actin filament models for ADP–F-actin and ADP-P$_i$–F-actin were generated using UCSF Chimera as previously described[41]. Starting from the modelled actin trimer, a copy was generated and the two terminal protomers on the barbed end were superimposed onto the two terminal protomers on the pointed end. These two protomers of the newly generated trimer were deleted, and the remaining protomer was combined with the original model, extending it by one protomer. This process was repeated iteratively until a 31-protomer filament was generated.

## Central pore and nucleotide cleft analysis

The CASTp web server[79] was used to identify continuous solvent-accessible pockets within the high-resolution helically symmetric F-actin structures. To eliminate boundary effects, each model was extended to five protomers as described above. Using an initial probe size of 1.4 Å revealed a solvent-accessible core that connected through narrow channels to the nucleotide pocket and broader channels to the filament's exterior and bulk solvent. Increasing the probe size to 1.6 Å isolated the central solvent channel from these pockets. Water molecules contained within this discrete pocket at the filament's core are displayed in Fig. 2a and Extended Data Fig. 2e.

The CASTp server was also used with a probe size of 1.4 Å to measure the volume of the solvent-accessible nucleotide pocket in the asymmetric, seven-protomer F-actin models. The nucleotide pocket volume of individual protomers from each model was measured to separate the nucleotide pocket from the filament's central core.

## Analytical modelling of thermal fluctuations

Boltzmann modelling of thermal bending fluctuations was performed as described in the Supplementary Discussion. Residuals between the modelled curves and experimental curvature histograms (Fig. 3b) were calculated by computing the difference between the modelled probability distribution and the histogram heights at the centre of each 0.2 μm$^{-1}$ bin. An adjusted Boltzmann distribution with a multiplicative constant $\alpha$ was fit to each dataset by minimizing the sum of the squared residuals between the adjusted model and the data, converging to $\alpha$ values of 0.80 for ADP–F-actin and 0.93 for ADP-P$_i$–F-actin. Model distributions featuring integer persistence lengths between 5 and 15 μm were additionally calculated to visualize the effects of varying persistence length on the curvature distribution.

## Rise, twist and curvature measurements

Rise and twist were measured along deformed actin filament axes using custom Python scripts, implementing an approach similar to previously described helix deformations[80–82]. First, a central axis was defined using a 3D spline fit. To minimize edge effects, the model was extended by copying the model twice, aligning the terminal three subunits of one copy's barbed end with the terminal three subunits of the original model's pointed end and then aligning the terminal three subunits

of the other copy's pointed end with the terminal three subunits of the original model's barbed end. The overlapping subunits from the copied models were deleted to generate a final, 42-protomer model. This was used to define the central spline while sufficiently minimizing edge effects.

To define the 3D spline for each filament's central axis, an iterative, orientation-independent approach was implemented using a set of waypoints. The initial 41 waypoints were defined as the set of 3D coordinates corresponding to the centroid of 2 consecutive subunits in a rolling window along the filament. A 3D cubic spline with a natural boundary condition was then fit through the set of waypoints to generate the initial filament axis. A set of line segments with a length equal to the filament's radius and one end positioned at each subunit's centroid was aligned to minimize the free end's distance from the spline. The waypoints were then updated to become the Euclidean average of two of these consecutive free ends. The process of updating the 3D cubic spline, defining new line segment extensions from the subunit centroids, and updating waypoints was repeated 500 times to obtain the final central axis spline.

Rise was measured by computing the distance travelled along the path of the central axis spline between protomer centroids. For each protomer, the point on the central axis that was the closest to the subunit's centroid was stored, and the distance along the spline path was calculated to the next protomer. The twist between protomers along the deformed short-pitch F-actin helix was measured in the context of the moving Frenet–Serret frame of reference. The Frenet–Serret frame of reference was defined by the orthonormal basis of the unit tangent, unit normal and unit binormal vectors along the length of the spline. The set of unit tangent vectors sampled at the positions along the 3D cubic spline corresponding to each subunit (as determined during the rise measurements) was calculated. A vector with a magnitude equal to the filament's radius oriented along the normal axis in the Frenet–Serret frame and its tail at the origin in the Frenet–Serret frame was then rotated in the normal-binormal plane until the distance between its head and the corresponding subunit centroid was minimized. This rotation angle defined the absolute angular twist for the protomer. To measure the twist along the short-pitch helix, the difference between consecutive absolute angular twists was calculated.

### Travelling-wave analytical model
Inspection of instantaneous twist versus protomer index plots for bent filaments revealed a clear sinusoidal pattern along each strand. Furthermore, as the curvature increased along the cryoDRGN trajectory, both the magnitude and position of this sinusoidal pattern changed. We therefore modelled the bend–twist phenomenon for each strand as travelling waves using the equation:

$$u(x, t) = Ax\sin(kx + \omega t + \varphi) + B,$$

where $u(x,t)$ is the instantaneous twist, $A$ is a coupling factor between curvature and twist amplitude, $k$ is the propagation factor that determines how rapidly the twist wave travels along the filament's length with bending, $x$ is the curvature, $\omega$ is the period of twist, $t$ is the position along the central axis (parameterized to protomer index), $\varphi$ is the phase shift of twist and $B$ is the overall average twist. This equation was jointly fit against the measured twist values and estimated curvatures for each of the 16-protomer cryoDRGN models. Curvature for each model was measured as the average of the instantaneous 3D curvatures of the central axis spline. For the ADP nucleotide state, all frames were used. For the ADP-$P_i$ state, the first three frames had low curvature and had fluctuating curvature measurements, so they were omitted due to the inaccurate average curvature measurement of the central spline. The fit values for the model parameters are presented in Supplementary Table 2, and example fit functions through experimental data are presented in Extended Data Fig. 6a,b.

### Analysing central axis deformations
Analysis of the bending deformations of filaments were performed on the central axes of the 16-protomer cryoDRGN models. Principal component analysis was performed on the coordinates sampled along the axis spline in Euclidean space. The plane defined by the first and second principal components represents the plane of maximum filament curvature. The plane formed by the first and third principal components represents an orthogonal plane capturing the 3D character of the bent filament. Central axis deformation was also analysed by measuring the deviation of the central axis from a straight line fit. For each curved cryoDRGN model, a straight line was aligned to the terminal 56 Å of its central axis at the barbed end. Discrete 0.28 Å sampling steps were then made along the central axis, and the distance from the sampled point and the straight line was plotted in Extended Data Fig. 5e.

### Actin subdomain measurements
Actin subdomains were defined using previously established residue assignment conventions[83]: subdomain 1 (SD1): amino acids 5–32, 70–144, 338–375; SD2: amino acids 33–69; SD3: amino acids 145–180, 270–337; SD4: amino acids 181–269. Using a custom Python script, the Euclidean distances, angles, and dihedral angles between subdomains indicated in Extended Data Fig. 6a were measured. The protomer indexing started at the pointed end and progressed to the barbed end. For measurements that spanned multiple protomers, the protomer index corresponded to the most pointed-end protomer.

### Subunit shear measurements
For shear measurements, each protomer of the asymmetric F-actin models was aligned to the protomer of the corresponding high-resolution, helically symmetric model of the same nucleotide state. The average displacement vector for each subdomain between these models was then computed. Observing anti-correlated displacements between non-adjacent subdomains led us to define two shear indices to describe these coordinated deformations: shear index 1, the dot product of the subdomain 1 and 4 displacement vectors, and shear index 2, the dot product of the subdomain 2 and 3 displacement vectors. Shear indices of pairs of subdomain displacement vectors that have large individual magnitudes and opposing directionality will have large negative values, indicative of shear, whereas small displacements or lack of correlated subdomain displacements will produce values near zero.

### Strain analysis
To quantify protein deformations not explained by rigid-body motions, strain pseudoenergy analysis was performed using a custom Python script, implementing a previously described approach[84,85]. In brief, a reference helical F-actin protomer was rigid-body fit into each protomer of the model to which it was being compared. The local deformation matrix within an 8 Å neighbourhood was estimated for each alpha-carbon of the reference protomer. The Eulerian strain tensor is computed using a first order approximation of the deformation matrix's spatial derivative. The shear strain energy is then calculated directly from this strain tensor. This approach to protein deformation has the major advantage of being rotationally invariant and distinguishing rigid-body motions from internal deformation. However, the local deformation estimation can be inaccurate for very large deformations, which limited our strain analysis to individual protomers. Furthermore, the first-order approximation assumes continuous, as opposed to granular, deformations, which makes the measurements relative pseudoenergies.

### Plots, statistics and molecular graphics
Plots were generated using GraphPad Prism or Matplotlib[86]. Statistical tests were performed using GraphPad Prism. Molecular graphics were prepared using UCSF Chimera[72] and UCSF ChimeraX[78].

## Reporting summary

Further information on research design is available in the Nature Research Reporting Summary linked to this article.

## Data availability

Cryo-EM density maps and corresponding atomic models have been deposited in the PDB and EMDB with the following accession codes: helically symmetric ADP–F-actin (PDB: 8D13, EMDB: EMD-27114); helically symmetric ADP-$P_i$–F-actin (PDB: 8D14, EMDB: EMD-27115); asymmetric bent ADP–F-actin (PDB: 8D15, EMDB: EMD-27116); asymmetric bent ADP-$P_i$–F-actin (PDB: 8D16, EMDB: EMD-27117); asymmetric straight ADP–F-actin control 1 (PDB: 8D17, EMDB: EMD-27118); asymmetric straight ADP–F-actin control 2 (PDB: 8D18, EMDB: EMD-27119). Cryo-EM datasets have been deposited in the EMPIAR with the following accession codes: ADP–F-actin (EMPIAR-11128); ADP-$P_i$–F-actin (EMPIAR-11129). These depositions include the raw movies and processed particle stacks used to generate the final reconstructions deposited in the EMDB. Datasets for cryoDRGN analysis, neural network training and cofilin severing assays are available at Zenodo. Synthetic datasets used to train denoising auto-encoder and semantic segmentation neural networks as well as the trained networks are available at https://doi.org/10.5281/zenodo.6917913. CryoDRGN reconstructions, fitted models, trained cryoDRGN networks and the data required to train the cryoDRGN networks are available at https://doi.org/10.5281/zenodo.6928604. Cofilin TIRF microscopy data are available at https://doi.org/10.5281/zenodo.6929148. All other data required to assess this study's conclusions are presented in the manuscript. Materials are available from the corresponding author without restriction. Source data are provided with this paper.

## Code availability

All custom code associated with this study is open source and is available for download without restriction. Cryo-EM analysis software is available at GitHub (https://github.com/alushinlab/bent_actin), and scripts for analysing TIRF videos are available at Zenodo (https://doi.org/10.5281/zenodo.6929148).

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

**Acknowledgements** We thank S. Espinosa de los Reyes for preparation of ADP–F-actin grids; R. Ranganathan for strain pseudoenergy calculation pseudocode; and H. Ng, J. Sotiris and M. Ebrahim for assistance with cryo-EM data collection at the Rockefeller University Cryo-electron Microscopy Resource Center. This work was funded by a National Institutes of Health grant (R01GM141044), an Irma T. Hirschl and Monique Weill-Caulier Research Award, and a Pew Biomedical Scholar Award to G.M.A. R.G. was supported by an H. Li Memorial Fellowship, and A.G.C. was supported by NIH T32 GM115327.

**Author contributions** M.J.R. and G.M.A. conceived the project. M.J.R. developed custom cryo-EM analysis software and performed all data analysis presented in the paper. C.H. prepared proteins, performed cofilin severing assays and prepared cryo-EM specimens. A.G.C. and R.G. prepared cryo-EM specimens and collected cryo-EM data. M.J.R. and G.M.A. wrote the paper, with input from all of the authors.

**Competing interests** The authors declare no competing interests.

**Additional information**
**Correspondence and requests for materials** should be addressed to Gregory M. Alushin.

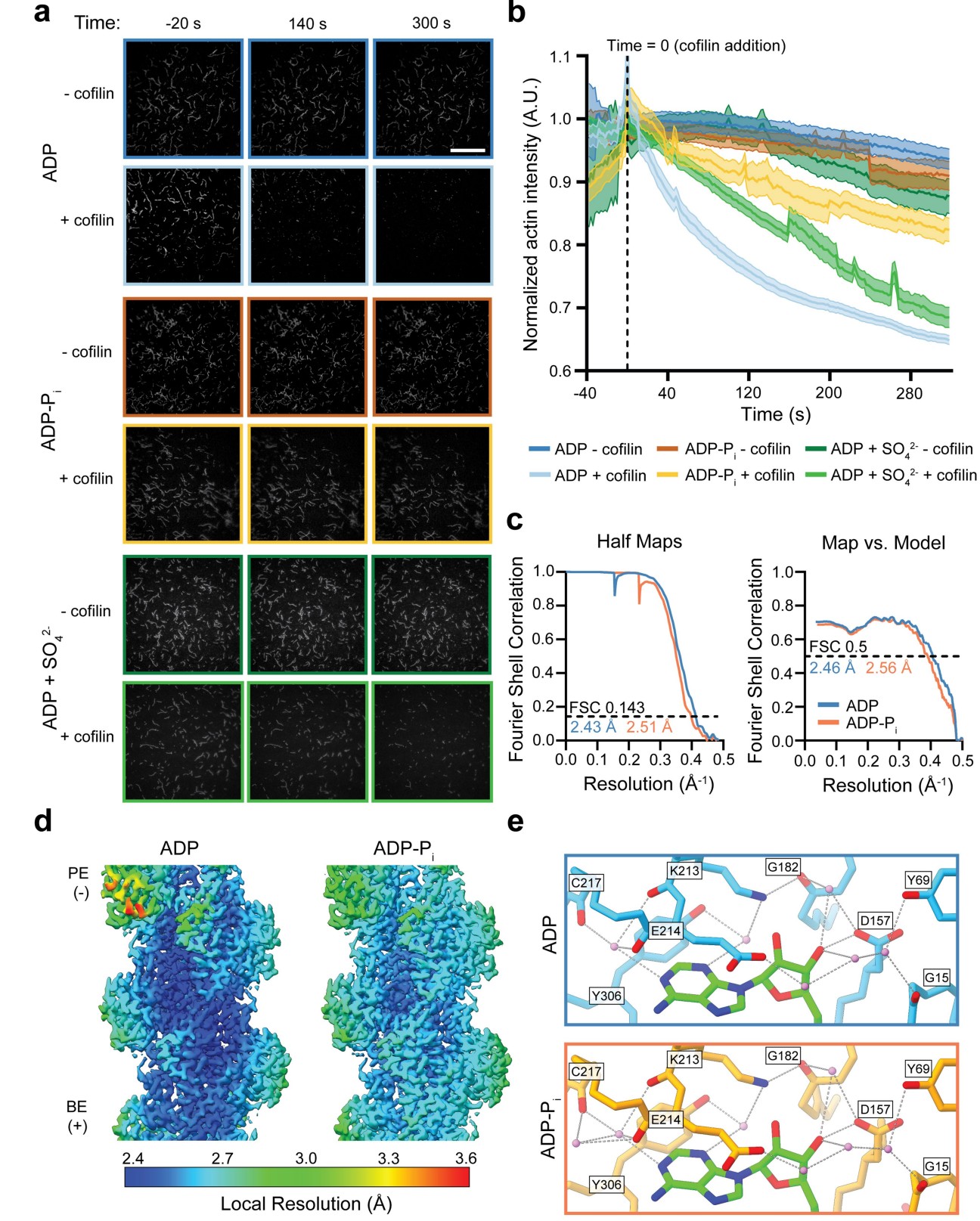

**Extended Data Fig. 1** | See next page for caption.

**Extended Data Fig. 1 | Validation of ADP–P$_i$-F-actin preparation and helically symmetric reconstructions. a**, Representative TIRF-microscopy video frames from cofilin severing assays. Cofilin-free controls are shown in the top row of each condition, indicated by a darker border. Scale bar, 40 μm. **b**, Quantification of TIRF videos showing the average normalized actin channel intensity. Error margin in graph indicates +/− 95% CI. Half-lives represent exponential decay at time 0 s, with 95% CI, and n values represent independent experiments: ADP–F-actin · cofilin (778 ± 24 s, n = 3); ADP-P$_i$–F-actin · cofilin (454 ± 14 s, n = 3); ADP-sulfate–F-actin · cofilin (348 ± 16 s, n = 3); ADP–F-actin + cofilin (50.4 ± 2.1 s, n = 4); ADP-P$_i$–F-actin + cofilin (177.5 ± 10.3 s, n = 3); ADP-sulfate–F-actin + cofilin (86.8 ± 2.5 s, n = 3). **c**, Half-map (left) and map-to-model (right) Fourier Shell Correlation (FSC) curves for helically symmetric reconstructions of ADP- and ADP-P$_i$-F-actin. **d**, Local resolution assessment of helically symmetric ADP–F-actin and ADP-P$_i$–F-actin. PE: pointed end; BE: barbed end. **e**, Potential hydrogen-bonding networks adjacent to the nucleosidyl region of ADP. Key side chains and back bone atoms participating in hydrogen-bonding networks are displayed and coloured by heteroatom.

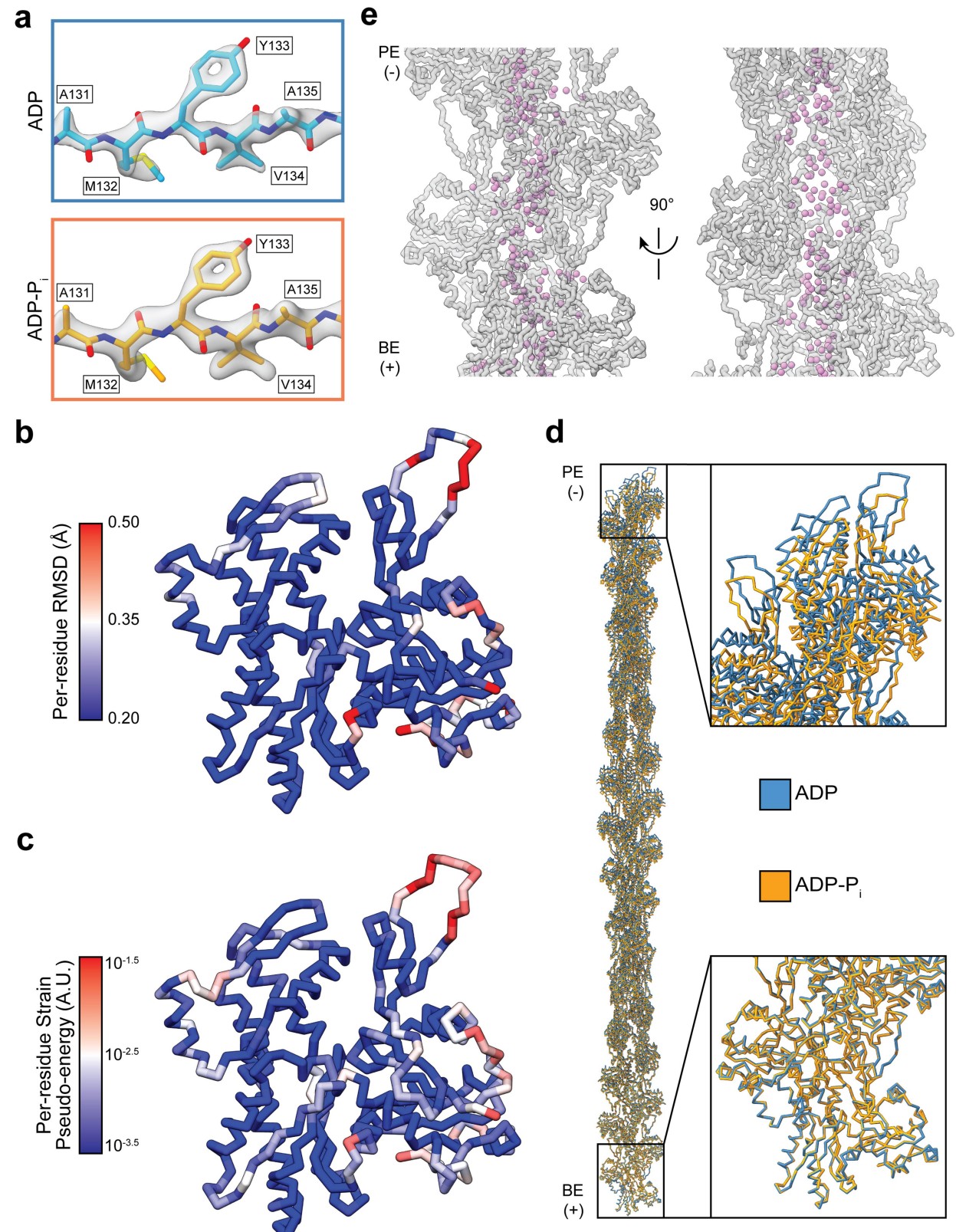

**Extended Data Fig. 2 | Additional analysis of helically symmetric ADP–F-actin and ADP-Pᵢ–F-actin models. a**, Example cryo-EM map density superimposed with atomic model residues A131–A135 from ADP–F-actin (top) and ADP–Pᵢ-F-actin (bottom). **b**, Individual ADP–F-actin protomer shown in $C_\alpha$ representation, coloured by per-residue RMSD between ADP–F-actin and ADP-Pᵢ–F-actin. **c**, Same as **b**, but coloured by per-residue strain pseudo-energy.

**d**, Superposition of extended 31-protomer ADP- (blue) and ADP-Pᵢ-F-actin (orange) models, aligned at the terminal barbed end protomer. **e**, Water molecules (violet) contained within the ADP-Pᵢ–F-actin filament's core. Actin subunits are shown in transparent grey backbone representation. PE: pointed end; BE: barbed end.

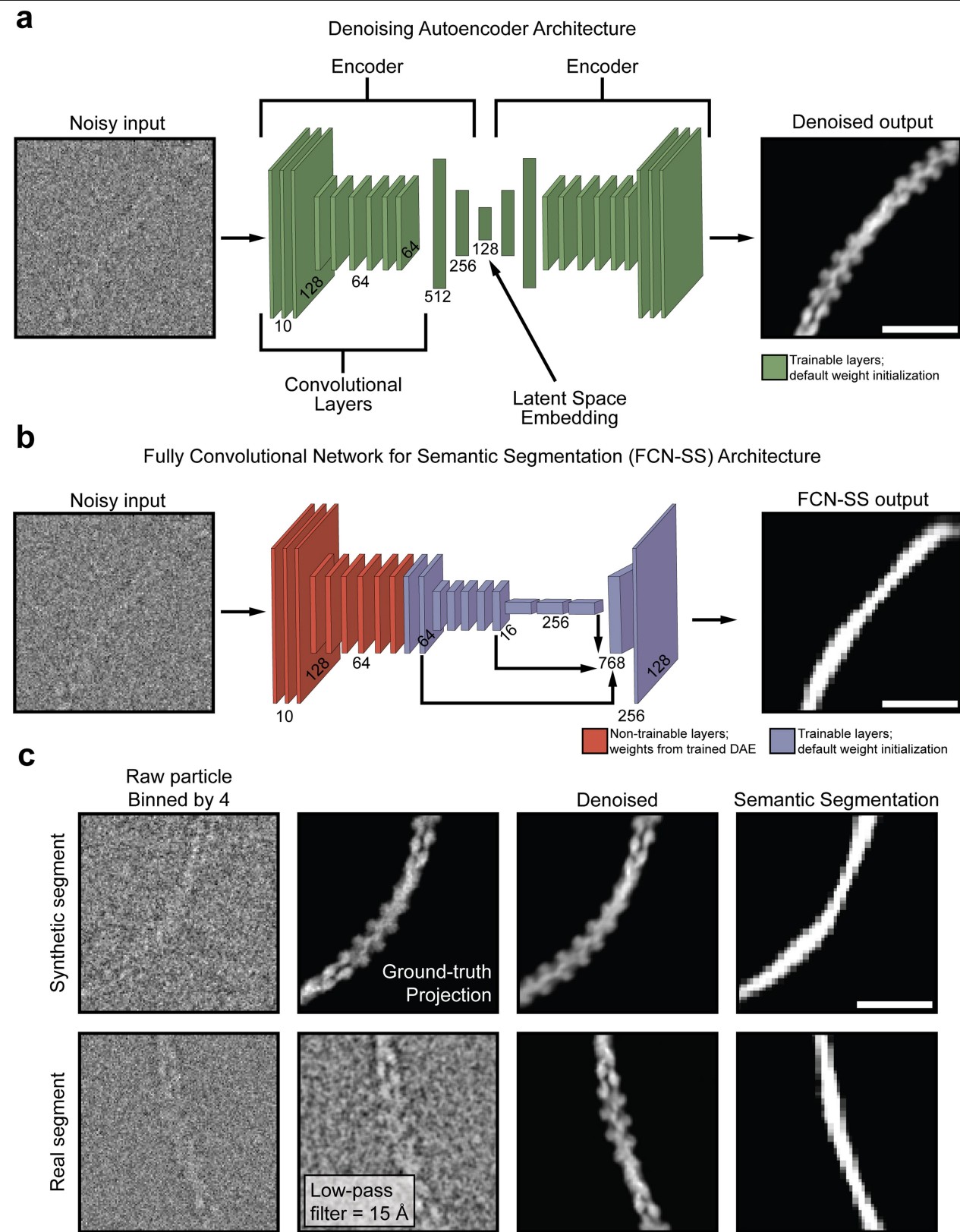

**Extended Data Fig. 3 | Neural network architecture and example performance. a**, Neural network architecture diagram for denoising auto-encoder. Example network input and output is displayed for a representative extracted segment. **b**, Network architecture diagram for semantic segmentation fully convolutional network. Example input and output for a representative extracted segment is shown. **c**, Representative network performance on filament segments from synthetic projections (top) and experimental cryo-EM micrographs (bottom). Scale bars, 20 nm.

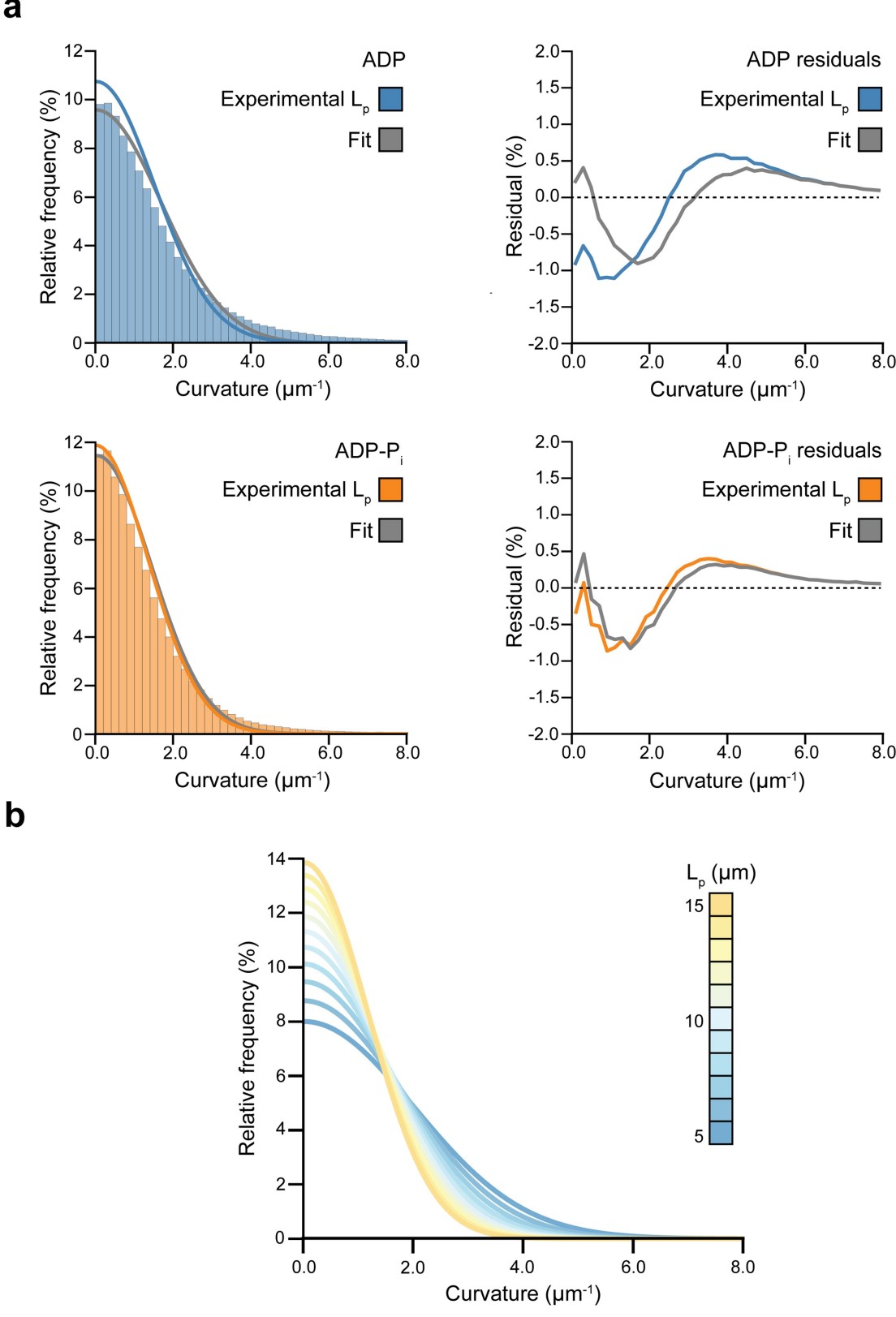

**Extended Data Fig. 4 | Boltzmann modelling of filament segment curvature distributions. a**, Curvature distributions (left column) of ADP–F-actin (top) and ADP–P$_i$-F-actin (bottom) with modelled thermal bending probability distributions. Residual plots (right column) between the measured and theoretical distributions. Grey curves correspond to adjusted bending models fit with a multiplicative parameter in the energy term. **b**, Theoretical probability distributions of 500 Å elastic rods bending due to thermal fluctuations, modelled as Boltzmann distributions. Varying the persistence length between 5 μm and 15 μm demonstrates the effect of filament bending stiffness.

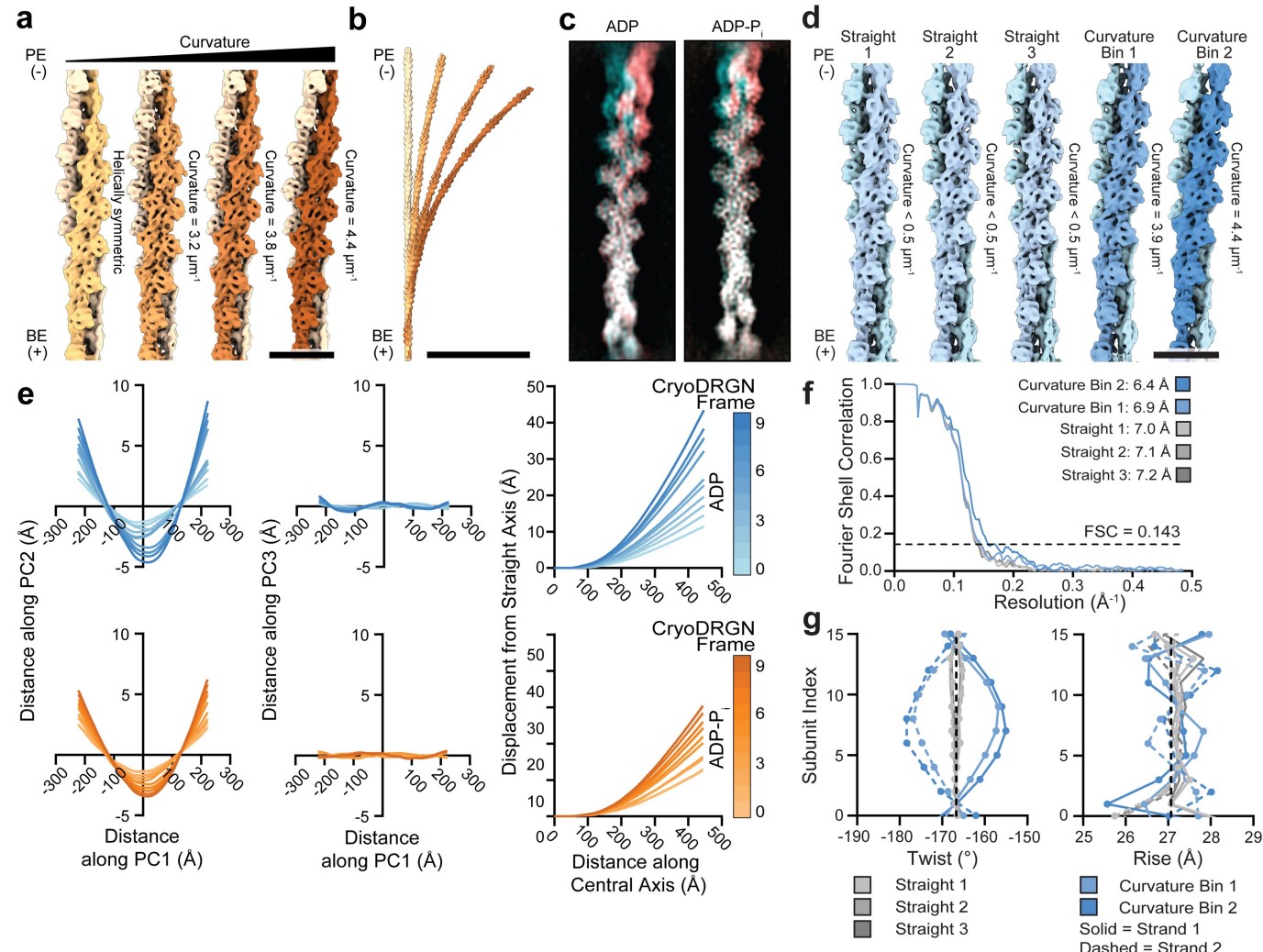

**Extended Data Fig. 5 | Additional analysis of filament bending deformations.**
**a**, Helically symmetric ADP-P$_i$–F-actin (left map) and cryoDRGN reconstructions
sampling ADP-P$_i$–F-actin bending (right three maps). Maps are lowpass filtered
to 8 Å, and strands are coloured in shades of orange. PE: pointed end; BE:
barbed end. Scale bar, 10 nm. **b**, Stitched volumes of straight and bent maps
from **a**, aligned on the bottom 16 protomers. Scale bar, 100 nm. **c**, Projections of
zeroth (cyan) and ninth (magenta) cryoDRGN reconstructions from ADP–F-actin
(left) and ADP-P$_i$–F-actin (right) aligned on the bottom protomer and oriented
to display maximum displacement. **d**, Asymmetric reconstructions of ADP–F-actin

from indicated curvature bins. Scale bar, 10 nm. **e**, Plots of central axis
deviations from straight lines in ADP–F-actin and ADP-P$_i$–F-actin cryoDRGN
reconstructions. First and second columns show principal component analysis
of the cryoDRGN reconstructions' central axes. Third column shows displacement
of the cryoDRGN reconstructions' central axes from straight lines which were
aligned to the barbed-end terminal 56 Å of the central axes. **f**, Half-map Fourier
Shell Correlation (FSC) curves for control asymmetric 16-protomer
reconstructions. **g**, Twist and rise measurements of control asymmetric
16-protomer reconstructions.

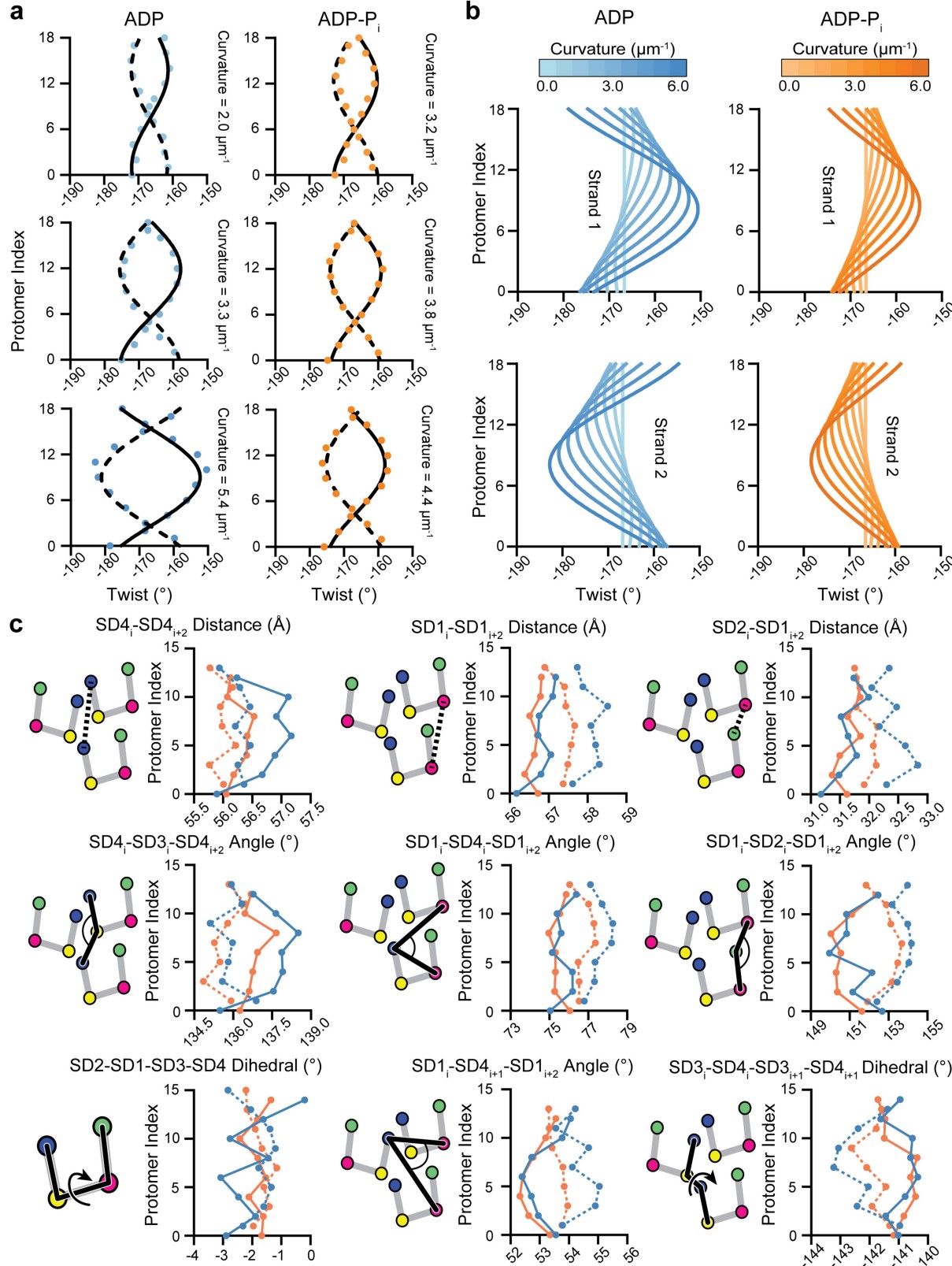

**Extended Data Fig. 6 | Quantitation of lattice architectural remodelling during filament bending. a**, Plots of travelling wave equation fits (black lines) with measured twist values (coloured points) from cryoDRGN reconstructions of different curvatures. **b**, Plots of twist travelling wave function at various curvatures, separated by strand. **c**, Plots of intra-strand, inter-strand, and intra-protomer subdomain distances and angles from ISOLDE models of the most curved cryoDRGN reconstructions of ADP–F-actin (blue) and ADP-P$_i$–F-actin (orange). Solid and dashed lines represent even (concave side) and odd (convex side) protomers, respectively.

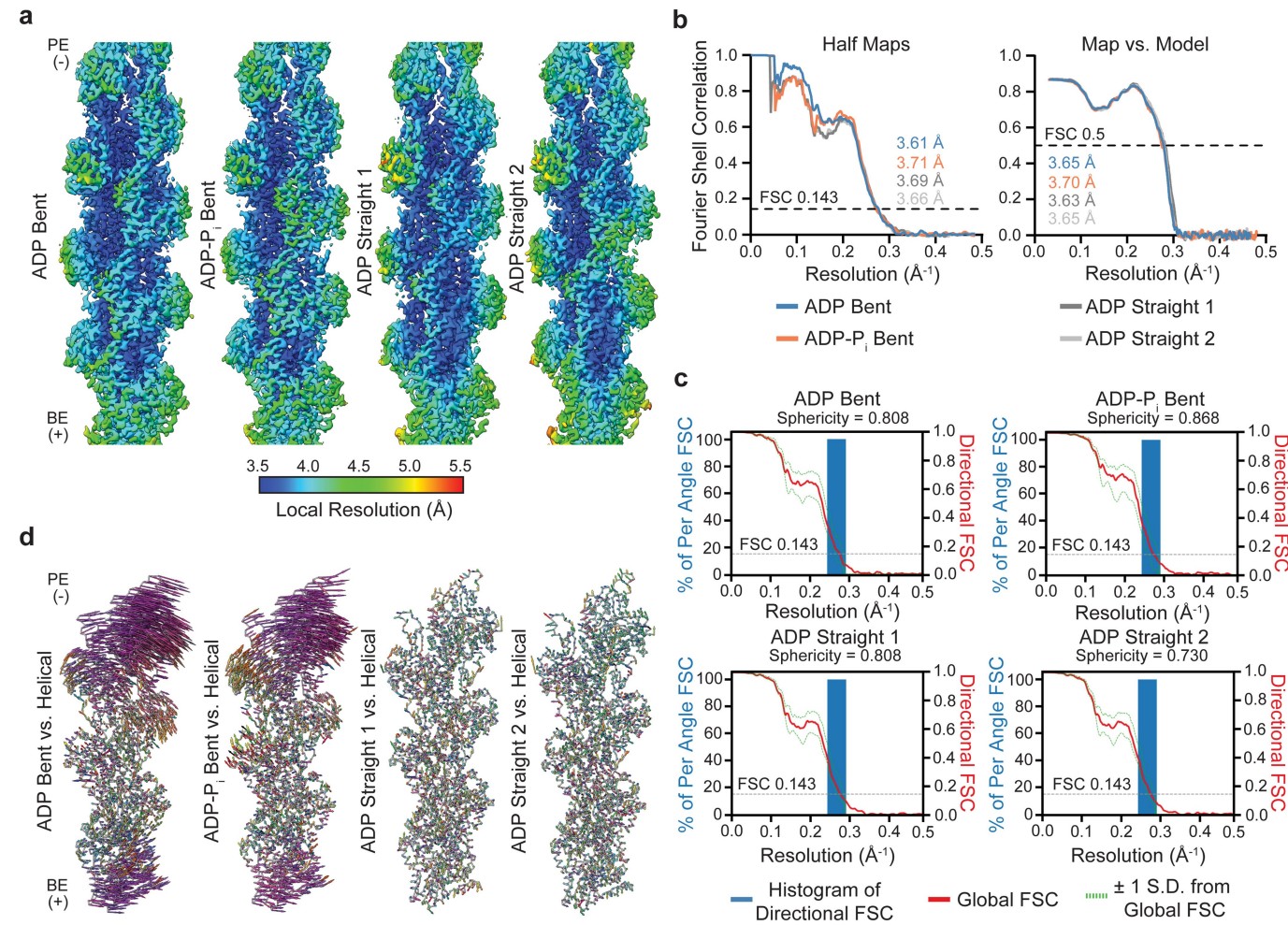

**Extended Data Fig. 7 | Resolution assessment and validation of high-resolution bent F-actin asymmetric reconstructions. a**, Local resolution assessment of bent ADP–F-actin and ADP-P$_i$–F-actin reconstructions, as well as two independent straight ADP–F-actin controls. PE: pointed end; BE: barbed end. **b**, Half-map (left) and map-to-model (right) Fourier Shell Correlation (FSC) curves for asymmetric bent ADP–F-actin, ADP-P$_i$–F-actin, and straight ADP–F-actin controls. **c**, 3D-FSC curves for asymmetric reconstructions, which indicate equivalently isotropic resolution between bent and straight reconstructions. Dotted green lines indicate +/− 1 s.d. from average FSC. **d**, Vector plots (coloured by direction and scaled 6X) representing C$_\alpha$ displacements between helically symmetric models and those built into indicated asymmetric reconstructions, aligned on the central protomer.

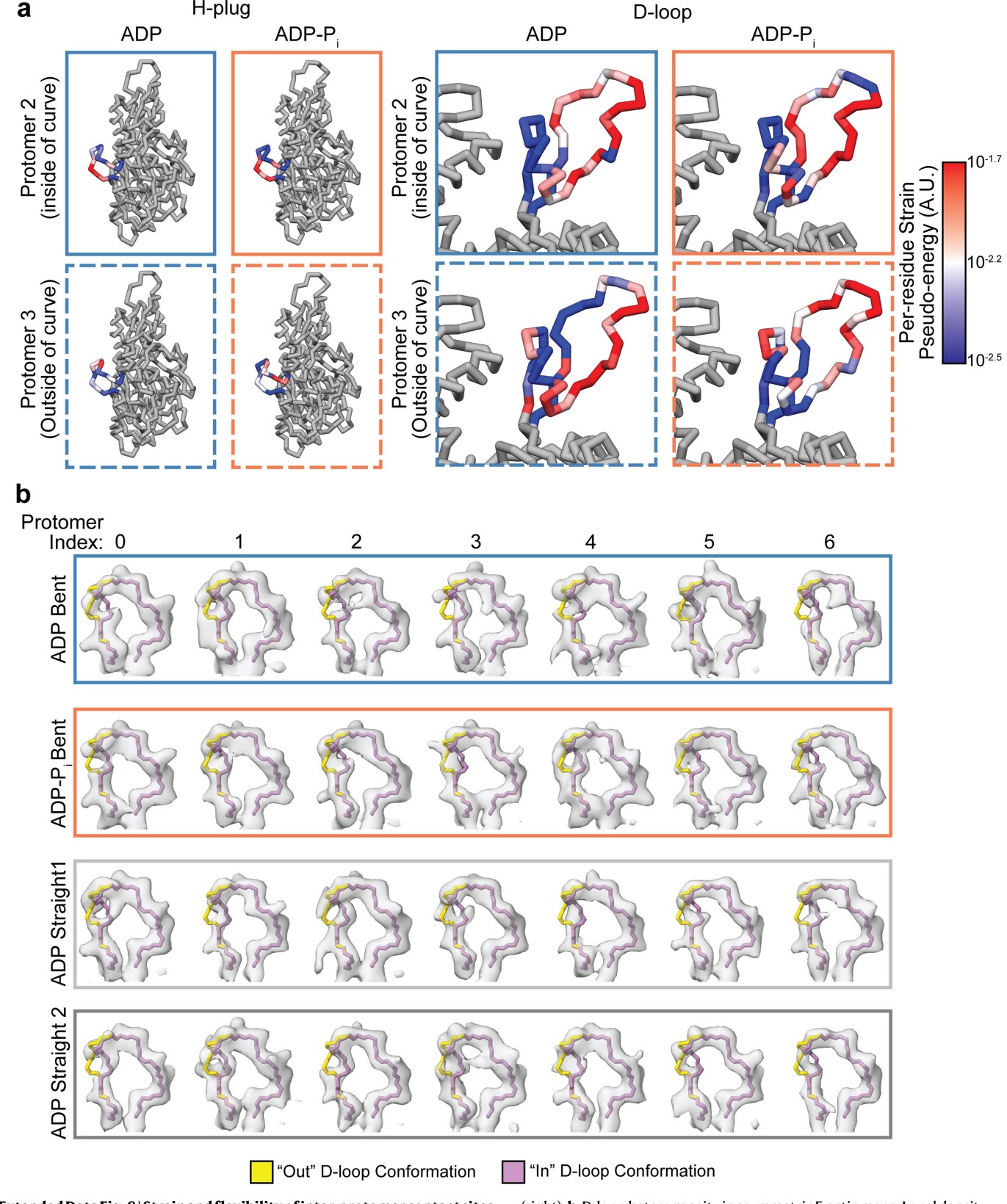

**Extended Data Fig. 8 | Strain and flexibility of inter-protomer contact sites in bent F-actin. a**, Grey $C_\alpha$ representations of indicated protomers from bent F-actin, locally coloured by computed per-residue strain pseudo-energy relative to helically symmetric models on their H-plugs (left) and D-loops (right). **b**, D-loop heterogeneity in asymmetric F-actin maps. Local density around each of the unique D-loops from the asymmetric reconstructions are shown with a docked PDB model of ADP–F-actin (7R8V) featuring both "in" and "out" D-loop conformations.

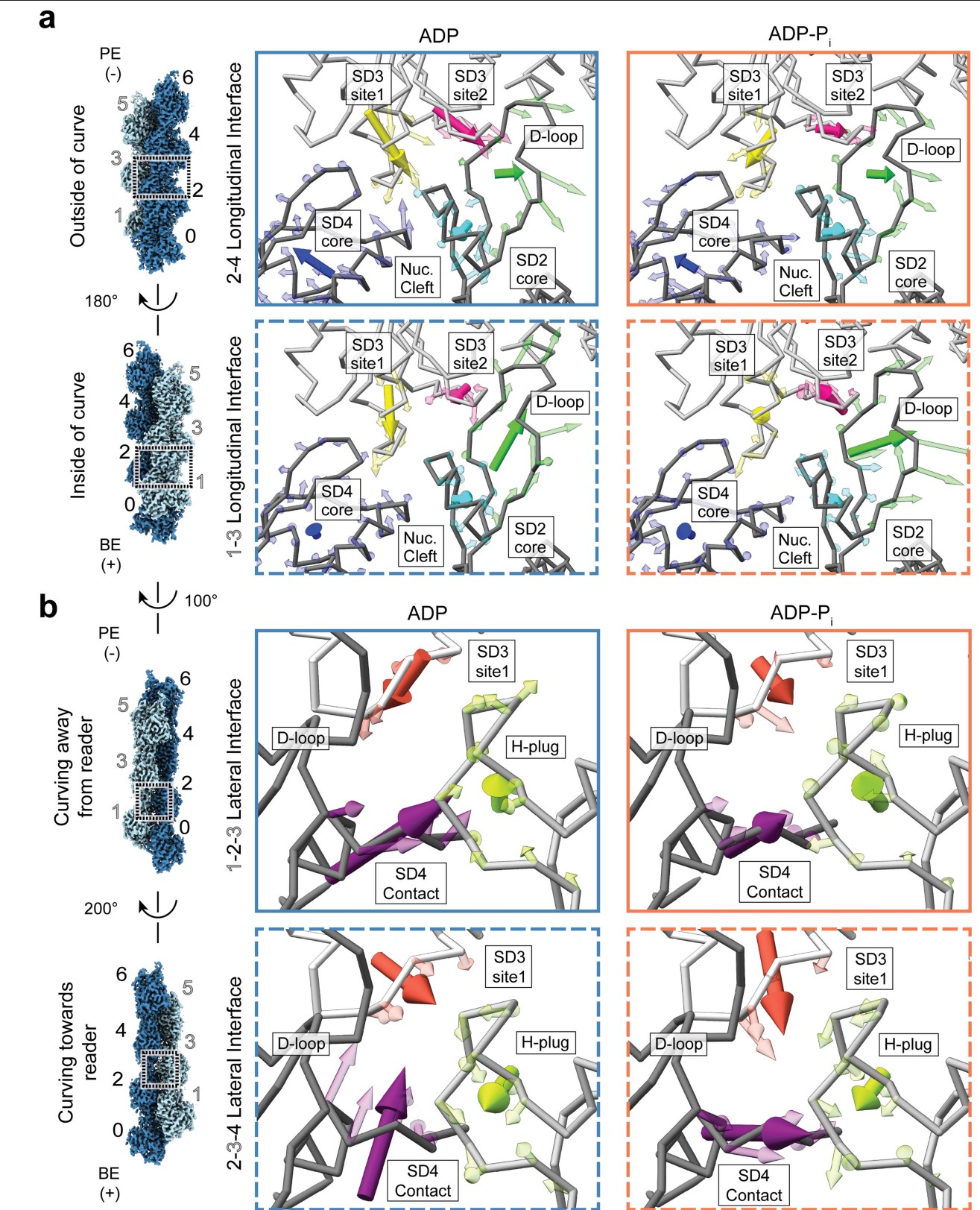

**Extended Data Fig. 9 | Steric encounters at inter-subunit interfaces transduce filament bending strain to protomers. a**, Longitudinal interfaces and **b**, lateral interfaces of bent ADP–F-actin (left) and ADP-P$_i$–F-actin (right). Solid and dashed borders represent inside (concave) and outside (convex) of curve, respectively. Transparent arrows represent individual C$_\alpha$ displacements from helically symmetric models scaled 6X, and solid arrows show averaged displacements of indicated regions scaled 20X. PE: pointed end; BE: barbed end.

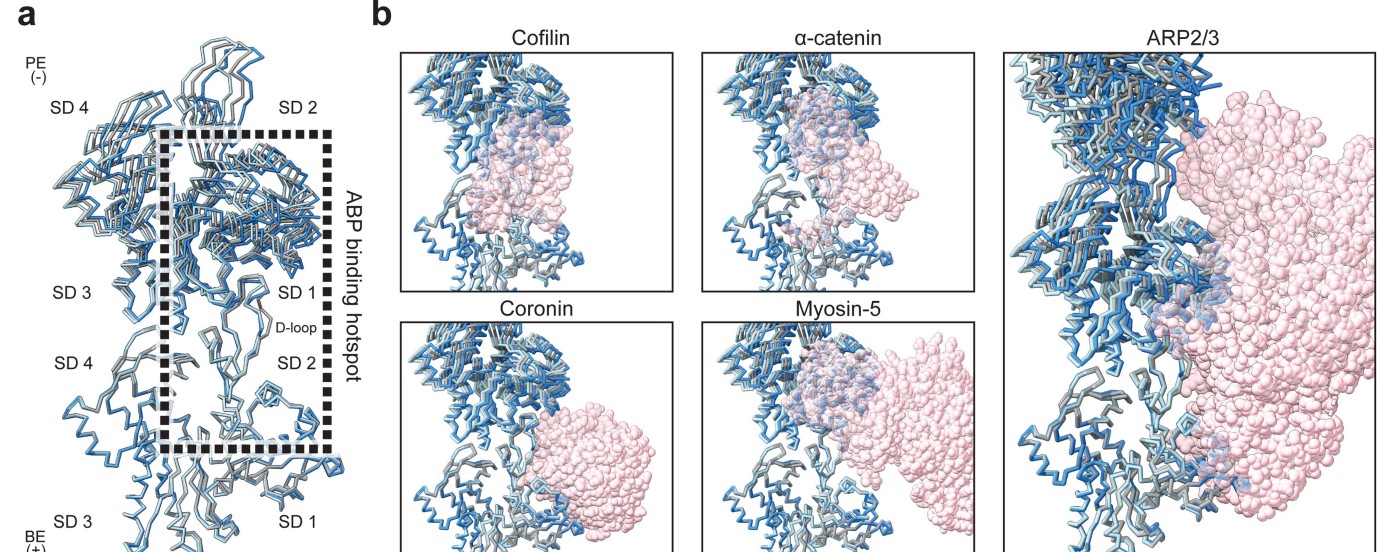

**Extended Data Fig. 10 | F-actin bending remodels inter-subunit interfaces engaged by ABPs. a**, C$_\alpha$ representations of inner strand protomer 6-protomer 8 (concave, cornflower blue) and outer strand protomer 7-protomer 9 (convex, light blue) longitudinal interfaces extracted from the most curved ADP–F-actin cryoDRGN 16-protomter atomic model, superimposed on the barbed end subunit of 2 protomers extracted from the ADP–F-actin helical model (grey).

Box encloses a region featuring major contacts by ABPs. BE: barbed end; PE: pointed end. Actin subdomains are indicated. **b**, Binding interfaces of indicated ABPs (pink space filling model) are displayed superimposed on the bent interfaces from **a**. Cofilin, PDB 5YU8; coronin, EMDB 6100 / PDB 2AQ5; ARP2/3, PDB 7TPT; α-catenin, PDB 6UPV; myosin-5, PDB 7PLT.

# Reporting Summary

## Statistics

For all statistical analyses, confirm that the following items are present in the figure legend, table legend, main text, or Methods section.

| n/a | Confirmed | |
|---|---|---|
| ☐ | ☒ | The exact sample size (*n*) for each experimental group/condition, given as a discrete number and unit of measurement |
| ☒ | ☐ | A statement on whether measurements were taken from distinct samples or whether the same sample was measured repeatedly |
| ☐ | ☒ | The statistical test(s) used AND whether they are one- or two-sided<br>*Only common tests should be described solely by name; describe more complex techniques in the Methods section.* |
| ☒ | ☐ | A description of all covariates tested |
| ☐ | ☒ | A description of any assumptions or corrections, such as tests of normality and adjustment for multiple comparisons |
| ☐ | ☒ | A full description of the statistical parameters including central tendency (e.g. means) or other basic estimates (e.g. regression coefficient) AND variation (e.g. standard deviation) or associated estimates of uncertainty (e.g. confidence intervals) |
| ☐ | ☒ | For null hypothesis testing, the test statistic (e.g. *F*, *t*, *r*) with confidence intervals, effect sizes, degrees of freedom and *P* value noted<br>*Give P values as exact values whenever suitable.* |
| ☒ | ☐ | For Bayesian analysis, information on the choice of priors and Markov chain Monte Carlo settings |
| ☒ | ☐ | For hierarchical and complex designs, identification of the appropriate level for tests and full reporting of outcomes |
| ☒ | ☐ | Estimates of effect sizes (e.g. Cohen's *d*, Pearson's *r*), indicating how they were calculated |

*Our web collection on statistics for biologists contains articles on many of the points above.*

## Software and code

Policy information about availability of computer code

| Data collection | SerialEM 3.7.0, Nikon NIS-Elements AR 5.21.03 |
|---|---|
| Data analysis | Custom machine learning software for cryo-EM analysis: https://github.com/alushinlab/bent_actin; python scripts for analyzing TIRF movies: https://doi.org/10.5281/zenodo.6929148.  CryoSPARC 2.11.0, EMAN2.22, RELION-3.1, RELION-3.0, UCSF Chimera 1.12, UCSF ChimeraX 1.2.4, python 2.7.14, TensorFlow 1.5.0, scikit-image 0.14.2, cryoDRGN 0.3.1, ISOLDE 1.2.0, COOT 0.9.1, Phenix 1.19.2-4158, resample.exe from FREALIGN v9.11, MotionCor2 1.3.2, CTFFIND 4.1.5, CASTp 3.0, Matplotlib 2.2.3, GraphPad Prism 9.4.1 |

For manuscripts utilizing custom algorithms or software that are central to the research but not yet described in published literature, software must be made available to editors and reviewers. We strongly encourage code deposition in a community repository (e.g. GitHub). See the Nature Portfolio guidelines for submitting code & software for further information.

## Data

Policy information about availability of data

All manuscripts must include a data availability statement. This statement should provide the following information, where applicable:
- Accession codes, unique identifiers, or web links for publicly available datasets
- A description of any restrictions on data availability
- For clinical datasets or third party data, please ensure that the statement adheres to our policy

Cryo-EM density maps and corresponding atomic models have been deposited in the PDB and EMDB with the following accession codes: Helically-symmetric ADP-F-

actin (PDB: 8D13, EMDB: EMD-27114); helically-symmetric ADP-Pi-F-actin (PDB: 8D14, EMDB: EMD-27115); asymmetric bent ADP-F-actin (PDB: 8D15, EMDB: EMD-27116); asymmetric bent ADP-Pi-F-actin (PDB: 8D16, EMDB: EMD-27117); asymmetric straight ADP-F-actin control 1 (PDB: 8D17, EMDB: EMD-27118); asymmetric straight ADP-F-actin control 2 (PDB: 8D18, EMDB: EMD-27119). Cryo-EM datasets have been deposited in the EMPIAR with the following accession codes: ADP-F-actin (EMPIAR-11128); ADP-Pi-F-actin (EMPIAR-11129). These depositions include the raw movies and processed particle stacks used to generate the final reconstructions deposited in the EMDB. Datasets for cryoDRGN analysis, neural network training, and cofilin severing assays are available through Zenodo. Synthetic datasets used to train denoising auto-encoder and semantic segmentation neural networks as well as the trained networks are accessible at https://doi.org/10.5281/zenodo.6917913. CryoDRGN reconstructions, fitted models, trained cryoDRGN networks, and the data required to train the cryoDRGN networks are accessible at https://doi.org/10.5281/zenodo.6928604. Cofilin TIRF microscopy data are accessible at https://doi.org/10.5281/zenodo.6929148. Source data are provided with this paper. All other data required to assess this study's conclusions are presented in the manuscript.

# Human research participants

Policy information about studies involving human research participants and Sex and Gender in Research.

| Reporting on sex and gender | N/A |
| --- | --- |
| Population characteristics | N/A |
| Recruitment | N/A |
| Ethics oversight | N/A |

Note that full information on the approval of the study protocol must also be provided in the manuscript.

# Field-specific reporting

Please select the one below that is the best fit for your research. If you are not sure, read the appropriate sections before making your selection.

☒ Life sciences  ☐ Behavioural & social sciences  ☐ Ecological, evolutionary & environmental sciences

For a reference copy of the document with all sections, see nature.com/documents/nr-reporting-summary-flat.pdf

# Life sciences study design

All studies must disclose on these points even when the disclosure is negative.

| Sample size | Sample sizes were not predetermined in our experimental design. TIRF assays were performed 3-4 times (exact n indicated in Extended Data Fig. 1 legend), as is standard practice for in vitro biochemical assays, with consistent results. For cryo-EM experiments, thousands of micrographs and hundreds of thousands of filament segments were analyzed (see Methods and Supplementary Table 1). The minimal sample size was determined to be that which resulted in consistent, high-resolution reconstructions. Multiple independently analyzed random subsets from each condition converged to the same three-dimensional structure, confirming this was achieved. |
| --- | --- |
| Data exclusions | No data were excluded. |
| Replication | 3-4 independent TIRF assays were performed for each condition (exact n indicated in Extended Data Fig. 1 legend). Cryo-EM datasets were collected in single multi-day sessions from individual specimens, as is customary in our field, and thus they were not directly replicated. However, the atomistic structural similarity between independently prepared and imaged ADP- and ADP-Pi-F-actin (Fig. 1) supports the reproducibility of our structure determination pipeline. |
| Randomization | For cryo-EM structural analysis, particles were randomly assigned to independent half datasets for resolution analysis. Randomization was otherwise not relevant for our study, which did not employ live animals or human subjects. |
| Blinding | Blinding was not relevant for our study, as it did not employ live animals or human subjects. |

# Reporting for specific materials, systems and methods

We require information from authors about some types of materials, experimental systems and methods used in many studies. Here, indicate whether each material, system or method listed is relevant to your study. If you are not sure if a list item applies to your research, read the appropriate section before selecting a response.

## Materials & experimental systems

| n/a | Involved in the study |
|-----|----------------------|
| ☒ | Antibodies |
| ☒ | Eukaryotic cell lines |
| ☒ | Palaeontology and archaeology |
| ☒ | Animals and other organisms |
| ☒ | Clinical data |
| ☒ | Dual use research of concern |

## Methods

| n/a | Involved in the study |
|-----|----------------------|
| ☒ | ChIP-seq |
| ☒ | Flow cytometry |
| ☒ | MRI-based neuroimaging |

