## [Peer Review File · Nature]

Manuscript Title: Bending forces and nucleotide state jointly regulate F-actin structure

Reviewer Comments & Author Rebuttals

Reviewer Reports on the Initial Version:

Referees' comments:

Referee #1 (Remarks to the Author):

Review: Reynolds et al., "Actin nucleotide state modules the F-actin structural landscape evoked by bending forces", Nature manuscript #2022-06-08690 (2022)

The authors present cryo-EM reconstructions of straight actin filaments (F-actin) in the ADP and ADP-Pi states at 2.4 - 2.5 Å resolutions. They were able to identify many water molecules in the nucleotide binding pocket, in the core of the double-stranded filament and in the subunit interfaces. The resolutions obtained are lower than those in a recent study by Oosterheert et al, bioRxiv 2022, which also presents a tri-phosphate structure. It is re-assuring that the two studies, presumably performed independently, come to the same conclusions: the F-actin structures in the different nucleotide states are remarkably similar, probably to within the experimental error in most parts of the structures.

However, Reynolds et al then come to a different conclusion: they consider it unlikely that the small number of significant differences observed explain that several actin binding proteins (ABPs) bind to F-actin differently depending on nucleotide state. Instead, the authors found that their careful analysis of bent filament structures, obtained using advanced techniques revealed another difference between ADP and ADP-Pi F-actin: ADP F-actin is more bendable. The difference is small (1.14 1/μm vs 0.96 1/μm) and the discussion of the various structural differences between the two bent filaments deals with small effects.

The analysis of the bent filaments leads the authors to agree with a previously proposed coupling of filament bending and twist changes, and to propose a "steric boundaries model" (Figure 6b) that describes how the filament, and its subunits accommodate the bending strain. There is also some impressive mathematical modelling of the filament architecture changes upon bending, which I enjoyed.

As far as I understand, the authors attempted to answer two questions: what happens when actin filaments bend? And, how do actin binding proteins sense nucleotide state, in particular the longer-lived ADP-Pi and ADP states? To me, the first questions is quite well answered by the present work, whereas the second is possibly not. The idea put forward here is that ABPs sense nucleotide state of the filament because it likes to bend more when in the ADP state. This idea is not well developed and is just that, an idea. Given the very small changes involved (see Figure 6a and all movies), to me at least it is equally unlikely as the other model put forward, that the straight filaments are different in the ADP and ADP-Pi states.

The work has been done with great care and the experiments have been performed well and to a very high technical standard. The manuscript is exceptionally well illustrated and is well written, although some of the description of structural changes is lengthy and at times a bit hard to follow (even for a structural biologist).

In summary, this is an extremely well-performed and presented study that analyses for the first time, at the atomic level, how actin filaments bend and how nucleotide state influences this process. The idea that a difference in bending propensity is the cause for differential ABP behaviour I find speculative, but interesting - it will need more work to be sure.

Some specific comments in no particular order:

1) abstract: "We therefore propose actin nucleotide state serves as a co-regulator of F-actin mechanical regulation, as bending forces evoke nucleotide-state dependent conformational transitions that could be readily detected by ABPs" does require that ABPs are dependent on there being bent filaments. How can we know that? ABPs play no role away from the membrane where there are presumably fewer bending forces? Is the 200 nm bending length scale in cells mentioned in the Introduction realistic when looking at the curvatures used in the study?

2) line 59: Does actin hydrolyse ATP as G-actin is incorporated into filaments? I would suggest that ATP-bound actin polymerises, the binding of the filament changes the conformation ("cytomotive switch"; G-F transition; flattening), the new conformation allows hydrolysis. It is correctly described in line 74.

3) line 71: Is it possible that there is something fundamentally wrong with the literature certainty that ABPs sense nucleotide state? Maybe those experiments were not done correctly? If the mechanical properties of filaments are different in different states, then the assays might have measured something different? Wording could take that into account.

4) line 85: Why is it unclear? Can ABPs not simply reach the nucleotide (see also comment about Figure 6a)?

5) lines 92-97: I am not sure I understand this statement. Cofilin binding changes the actin filament. The fact that different nucleotide states give the same change (final state) does not mean that they all show the same kinetics and energies of getting there? See also 11)

6) line 123: "High salt": it is only 15 mM, correct? Not sure that is high salt. Also: how much Pi is in cells? Does ADP F-actin exist in cells? (Sorry, I am not an actin expert ...).

7) line 126: Not sure two digits needed in cryo-EM resolutions, since FSC estimates are not very precise.

8) line 130: How were water molecules identified? Were they manually assigned and also checked for chemical plausibility (an old and solved problem in crystallography)?

9) line 135: Does the comparison to the Raunser study (Oosterheert et al, bioRxiv 2022) also extend to all/important water molecules?

10) line 139: Such small RMSD differences will be dominated by the refinement protocol used. Maybe mention this? The number is basically meaningless and does not represent the experiment.

11) line 146-149: Again, I am not sure this is right: cofilin binding will introduce large conformational differences, but this uses the energy of its binding, and that energy might be in surplus for at least one form. What the energetics or kinetics of such a change are is not necessarily obvious in structures of the starting and final states (which are very similar here in terms of structure (+/- Pi), but it can be the transition state that makes the difference.

12) lines 176-177: The interfaces have to be solvated because the subunits (G-actin) need to be soluble in the cytoplasm. This is in contrast to largely hydrophobic interfaces that never or rarely come apart. This is not a surprise or unusual.

13) line 192: "Significantly": how can we be sure this is significant? No estimation of errors given or variances? Line 203: variabilities do seem different, making averages possibly problematic?

14) lines 204-205: Resolution could be estimated with FSCs against atomic models. 8 Å seems quite low. Not possible to collect a really big dataset and classify one particular curvature? Ah. I see, this has been done - lines 252-. So, do we still need the cryoDRGN analysis for this study? Why not only use the higher-res data?

15) line 234: "Inter-strand angles also separated by reference strand," – something wrong with this sentence, I think?

16) lines 239-242: Thank you for pointing this out! (cytomotive switch; G-F transition; flattening is different from bending deformations!

17) lines 246-247: Resolutions of those more "traditional" reconstructions similar? Give numbers here?

18) lines 252-332: Quite a number of structural and deformation details are listed here, describing very small differences. Some "active"/colourful language used (pressing down etc) that makes looking at the effects disappointing, at least to me.

19) lines 345-346: That's not very clear to me: what does "structural landscape" mean here and how would ABPs work with this? ABPs would then basically just discriminate more-or-less straight/bent filaments? Is that what is meant and is that meaningful in a cell (I don't know)?

20) line 354/364: "Steric boundaries" model - first time this term is introduced here as a conceptual advance. Mention earlier what it is? Needs its own paragraph? Refer to Fig. 6b here.

21) Something that could help in the future: if cofilin is added to ADP and ADP-Pi F-actin filaments

(under non-saturating conditions), what happens then? Are differences observable by cryo-EM in terms of how many bind to what nucleotide state/curvature?

22) Figures are very well done and very clear!

23) Figure 1C: For non-experts, label subdomains (two nomenclatures), D-loop, H-loop, nucleotide binding pocket etc

24) Figure 4e: Why is there so much scatter in the AU units that seems non-random?

25) Figure 6a: How do the ABPs mentioned bind precisely? Supp figure might be useful to understand how such small differences could lead to differential effects? Also: the “ABP binding box” includes two nucleotide binding pockets. Can these ABPs really not sense nucleotide much more directly? (Sorry, don't know).

25) Figure 6b is important for the manuscript. Move its reference out of discussion / have a chapter summarising the model/mechanism?

26) Figure 6b: a brief explanation what is depicted in 6b would have helped me a lot to understand quicker. 6b is not referenced in the text, I think?

27) Supp Table 1: Why did ADP-Pi need twice as many particles to reach a lower resolution?

Referee #2 (Remarks to the Author):

In this work, Reynolds and coworkers use cryo-electron microscopy to solve new structures of filaments actin in ADP and ADP+free phosphate states. These structures allow them to detail possible roles for hydrogen bonded solvent in mediating interactions between actin subunits. Then, developing and training a machine learning algorithm to classify actin filament segments by curvature, the authors solve the structure of actin filaments for various amounts of bending. These structures contain as yet previously un-measured changes subunit structure (nucleotide dependent) that have implications for affinity of actin-binding proteins and, overall, the mechanical regulation of the actin cytoskeleton. The latter results, and in particular, the methodology leading to them, are completely novel to my understanding. Therefore, the results of this manuscript have the potential to make a large impact both on our understanding of stressed actin filaments, as well as the way in which researchers process their EM data. Below, I have a few comments and questions that I hope will help improve the manuscript.

1) Because a substantial part of the manuscript is about the role of bound waters, it would be helpful to compare these results to prior studies discussing the role of water in the nucleotide pocket. In particular, this paper from Voth (10.1016/j.jmb.2011.07.068) is a detailed study of this topic. But, perhaps that study is now out of date based on these new data, which would be very helpful to know. It would also be good to know how these new data compare to detailed studies on the actual

hydrolysis process and the final state, if possible (10.1021/acs.jctc.7b00077).

2) I'm unclear as to the resolving of D-loop conformations in the various structures. The text and supplemental figure 2 discuss the variability in the D-loop. However, Fig1C to me looks like the D-loops are superimposed almost exactly. Is that because it was modeled in using the same dihedrals despite uncertainty in the electron density? Perhaps this could be clarified further.

3) For water bridging interactions between subunits, I'm curious if the authors could make a comparison and statement about the possible role of divalent cations in bridging the subunits at these locations, as described in 2-3 papers by Enrique De La Cruz and coworkers. Would inclusion of specifically bound cations at these locations be able to be accommodated in these new structures or would it require substantially different water mediated interactions?

4) Bent actin filaments

a) It would be helpful to give the curvature values in Fig 3c,d, and in other locations where curvature is illustrated (e.g. S4a,b)

b) How long are the filaments used to compute the structures of curved segments? It seems for solving the structure, always the same number of subunits/protomers are used (16), but is this structure then coming from many adjacent/overlapping regions along a curved region? 16 protomers is I guess $2.6 \times 16 \sim 42$ nm long [by the way, line 924 says 56 angstroms which is the longitudinal spacing between actin subunits as stated on line 927, but the text says this is the length of one promoter. I'm not sure if that's a typo or not].

The highly curved segments appear to be on the order of 50 nm long in Fig 3a, but the structures illustrating curvature in Fig 3d are perhaps 300nm long, so I'm not sure which is representative of most of the data.

c) Related, how long is the model filament used for training in terms of number of actin subunits? I presume a fixed length filament is bent in order to generate the training data.

Since 2d images of synthetic bent filaments are used based on these data, do these 2d images always have the filament centered in the middle of the image, or could it have just a small piece of the filament in the synthetic imaging area?

d) Finally and related, in figure 3b, what is the length of actin subunits that are histogrammed here. Are they all the same length at fixed radius of curvature? This relates to my next point.

e) A question I still have reading the manuscript is how likely are these curvatures to arise by thermal fluctuations, and how likely are they with a particular amount of shear force. I believe the paper would be strengthened with some back of the envelope calculations to this end.

As a thought experiment, I considered what would give rise to the distribution in Figure 3b. Roughly, the energy to bend a filament of length L to a curvature κ should be something like $\frac{1}{2} k_{\text{bend}} * L * \kappa^2 = \frac{1}{2} kT * L_p * L * \kappa^2$ where L_p is the persistence length (with perhaps a factor

of two wrong because it is bent in an arc and not a circle). The distribution of bends without force is Boltzmann distributed, $P(E) \sim e^{-E/kT}$, so in terms of bending, a half gaussian, $P(\kappa) \sim e^{-1/2 L \rho L \kappa^2}$ (and then normalized). If I assume a persistence length of 10 microns and a length of 50 nm, I get a distribute not unlike that in figure 3b. However, I don't know the proper length L to use as in my previous question. This also assumes the filament behaves like a single elastic rod and not a discrete molecular system.

Nevertheless, I think this distribution could be computed, and the data in fig 3b could actually be used to compute a p-value of whether this comes from thermal fluctuations as the null hypothesis.

Interestingly, the expected distribution depends strongly on length of filament when the x-axis is curvature, so I don't know if my assumptions are correct here.

There may also be a way to estimate what force would produce a distribution similar to what is seen, which probably does extend to larger curvatures than expected based on the previous argument, and this would be very helpful to know.

5) Additional data:

As far as I can tell, the current manuscript doesn't have the pDBs of the new models included (just the validation for submission), and the GitHub link for the software is not available yet. It would be nice to be able to quickly review these. Additionally, I suggest the authors include the synthetic training set for the neural network models of actin filaments bent by different curvatures, as that may be helpful for future researchers.

Author Rebuttals to Initial Comments:

We thank the reviewers for their highly positive and constructive comments, and their recognition of the overall significance of our findings. We hope they and the editors will now find our paper acceptable for publication in *Nature*.

While no major issues were identified, numerous minor issues were brought up that we have done our best to address through textual edits, as well as some new analysis / analytical modeling. However, we note that due to editorial requirements, we had to shorten our paper by approximately 40%, so we hope the reviewers understand that we had to be judicious in our changes. Additionally, some detailed analysis most likely to be of interest to a subset of specialized readers has been moved to a Supplementary Discussion, which we believe streamlines the presentation in the main text for the general reader.

The most notable addition is analytical modelling of the plausible force regimes, and the likely role of passive thermal fluctuations versus active fluid forces (e.g. from blotting), in generating the filament bending distribution we observe. We thank reviewer 2 for this suggestion, as we believe inclusion of this analysis, as well as a discussion of the caveats of associated assumptions, substantially enhances the paper.

A detailed point-by-point response follows.

Referee #1 (Remarks to the Author):

Review: Reynolds et al., “Actin nucleotide state modules the F-actin structural landscape evoked by bending forces”, Nature manuscript #2022-06-08690 (2022)

The authors present cryo-EM reconstructions of straight actin filaments (F-actin) in the ADP and ADP-Pi states at 2.4 - 2.5 Å resolutions. They were able to identify many water molecules in the nucleotide binding pocket, in the core of the double-stranded filament and in the subunit interfaces. The resolutions obtained are lower than those in a recent study by Oosterheert et al, bioRxiv 2022, which also presents a tri-phosphate structure. It is re-assuring that the two studies, presumably performed independently, come to the same conclusions: the F-actin structures in the different nucleotide states are remarkably similar, probably to within the experimental error in most parts of the structures.

Indeed, these studies were performed entirely independently, and we also find it gratifying that they come to highly similar conclusions (as noted in lines 114-115). The reviewer is correct that the quantitatively assessed resolution of our maps (FSC 0.143) is slightly lower than those of Oosterheert et al., but we believe the map quality and details are similar based on the figures presented by Oosterheert et al. in their pre-print.

However, Reynolds et al then come to a different conclusion: they consider it unlikely that the small number of significant differences observed explain that several actin binding proteins (ABPs) bind to F-actin differently depending on nucleotide state. Instead, the authors found that their careful analysis of bent filament structures, obtained using advanced techniques revealed another difference between ADP and ADP-Pi F-actin: ADP F-actin is more bendable. The difference is small (1.14 1/μm vs 0.96 1/μm) and the discussion of the various structural differences between the two bent filaments deals with small effects.

The analysis of the bent filaments leads the authors to agree with a previously proposed coupling of filament bending and twist changes, and to propose a “steric boundaries model” (Figure 6b) that describes how the filament, and its subunits accommodate the bending strain. There is also some impressive mathematical modelling of the filament architecture changes upon bending, which I enjoyed.

As far as I understand, the authors attempted to answer two questions: what happens when actin filaments bend? And, how do actin binding proteins sense nucleotide state, in particular the longer-lived ADP-Pi and ADP states? To me, the first questions is quite well answered by the present work, whereas the second is possibly not. The idea put forward here is that ABPs sense nucleotide state of the filament

because it likes to bend more when in the ADP state. This idea is not well developed and is just that, an idea. Given the very small changes involved (see Figure 6a and all movies), to me at least it is equally unlikely as the other model put forward, that the straight filaments are different in the ADP and ADP-Pi states.

The work has been done with great care and the experiments have been performed well and to a very high technical standard. The manuscript is exceptionally well illustrated and is well written, although some of the description of structural changes is lengthy and at times a bit hard to follow (even for a structural biologist).

In summary, this is an extremely well-performed and presented study that analyses for the first time, at the atomic level, how actin filaments bend and how nucleotide state influences this process. The idea that a difference in bending propensity is the cause for differential ABP behaviour I find speculative, but interesting - it will need more work to be sure.

We thank the reviewer for this balanced yet overall positive evaluation of our paper, which we believe is an accurate summary. The reviewer's main critique, that we do not conclusively demonstrate the small changes evoked by nucleotide state in canonical F-actin have no effect on ABP engagement, is well-taken. While we believe the protomer-level structural changes may be within the realm of atomic modelling error (discussed later), the 0.3 Å change in rise could formally be detected by ABPs. We have modified the Abstract (lines 34-36), and Discussion (lines 259-261), to capture this idea. Nevertheless, we do believe that the structural changes we detail upon bending, notably the remodeling of interfaces, is more likely to regulate ABP engagement, as we note. We agree that this will require follow-up studies beyond the scope of this paper to elucidate, and we hope our study will stimulate the field to pursue them.

Some specific comments in no particular order:

1) abstract: "We therefore propose actin nucleotide state serves as a co-regulator of F-actin mechanical regulation, as bending forces evoke nucleotide-state dependent conformational transitions that could be readily detected by ABPs" does require that ABPs are dependent on there being bent filaments. How can we know that? ABPs play no role away from the membrane where there are presumably fewer bending forces? Is the 200 nm bending length scale in cells mentioned in the Introduction realistic when looking at the curvatures used in the study?

We have modified this sentence to capture the idea that small changes in canonical F-actin could also be relevant (line 41). While we focus on bending forces at the membrane, where they are prominent and have an established physiological function, such forces are likely to be at play throughout the cell (e.g. due to myosin activity, a topic for separate studies).

We do believe the ~200 nm bending length scale mentioned in the introduction is relevant. Figure 3d shows end-to-end stitches of the ADP cryoDRGN maps, which we performed to illustrate the extent of bending captured by the cryoDRGN analysis at the 100s of nm length scale of actual actin filaments in the cell that readers may typically see from PREM/tomography data of the leading edge of a cell. While this is a "quick and dirty" analysis, we gratifyingly find that the average curvature values measured from the central axis of bent models matches that obtained from stitching filament segments into an arc. We realized we had inadvertently omitted discussing this figure panel in our initial submission, and this is now explicitly stated in the text (lines 173-175).

2) line 59: Does actin hydrolyse ATP as G-actin is incorporated into filaments? I would suggest that ATP-bound actin polymerises, the binding of the filament changes the conformation ("cytomotive switch"; G-F transition; flattening), the new conformation allows hydrolysis. It is correctly described in line 74.

We have adjusted this to the wording "concomitant" (line 58) to capture what the reviewer is saying, which is correct.

3) line 71: Is it possible that there is something fundamentally wrong with the literature certainty that ABPs sense nucleotide state? Maybe those experiments were not done correctly? If the mechanical properties of filaments are different in different states, then the assays might have measured something different? Wording could take that into account.

Candidly, we do believe some of the literature regarding actin nucleotide-state sensing may be problematic. However, we do not want to engage in an extended / polemic discussion of that here, both due to space considerations and as we believe it would be more productive in the context of a review article. We believe this perspective is implied by our statement that “protocols for producing ADP-Pi F-actin vary”, lines 98-99. However, we are confident in cofilin’s nucleotide-state sensing, which we have explicitly validated here (Extended Data Fig. 1). This is one of the reasons we focused much of our discussion on cofilin in the paper.

4) line 85: Why is it unclear? Can ABPs not simply reach the nucleotide (see also comment about Figure 6a)?

We thank the reviewer for pointing out this ambiguity. We have now included an explicit statement that the nucleotide pockets of actin subunits within F-actin are buried and inaccessible to ABPs (line 84): hence, the field’s prior invocation of an allosteric model.

5) lines 92-97: I am not sure I understand this statement. Cofilin binding changes the actin filament. The fact that different nucleotide states give the same change (final state) does not mean that they all show the same kinetics and energies of getting there? See also 11)

Here we respond additionally to point 11, which is similar as indicated by the reviewer.

We believe there are essentially three potential models, which are not completely mutually exclusive, to explain nucleotide state sensing by cofilin / other ABPs. For simplicity we focus on cofilin, which binds and severs ADP F-actin better than ADP-Pi-F-actin.

Model a) Phosphate dissociation evokes a conformational change in F-actin, producing a structural state in F-actin that is more similar to the cofilin-bound state than ADP-F-actin (classical allostery model). As the reviewer notes, cofilin’s engagement confers binding energy, and the cofilin-bound state is indeed structurally distinct from both unbound ADP-F-actin and unbound ADP-Pi-F-actin, featuring highly deformed actin subunits and modified helical architecture. From our understanding of the classical allostery model, which has been discussed in prior literature that is cited in the main text, the logic of this model is as follows: If the ADP state more closely resembles the cofilin-bound state, less energy will be required to further deform the filament, promoting cofilin binding. We believe our data do not particularly support this model.

Model b) Phosphate dissociation does not need to directly cause an F-actin conformational change. Instead, it changes the mechanical properties of F-actin, such that ADP-F-actin is more deformable than ADP-Pi-F-actin. In this model, which has been favored by Enrique de la Cruz and colleagues, less mechanical work is required to deform F-actin via the binding energy associated with cofilin engagement. Our data are fully compatible with this model, as noted in the Discussion (lines 264-266).

Model c) An extension combining a and b. While their ground states are identical, mechanical force applied to ADP-F-actin can cause it to locally adopt a conformation more similar to the cofilin-bound state (modified lattice twist), which in addition to its enhanced deformability could promote cofilin binding. We note that, in addition to ADP-Pi-F-actin being less deformable, the same amount of curvature produces less architectural remodeling of the helical lattice (lines 189-191). Thus, this model suggests Pi does more than simply stiffen F-actin: it modifies the structural landscape which can be detected (and further modified) by ABPs. Our data also supports this model, and it is the one most extensively discussed in the paper.

The reviewer is correct that the kinetic processes and structural pathways leading to the cofilin bound state are likely to be complex, and they are not delineated by the results we have presented in our paper. We are optimistic that future molecular dynamics simulations guided by the structures we present here, which can explicitly address these issues, will be able to dissect the dynamic mechanisms in detail. However, we believe that is beyond the scope of the present study. While we hope to give these mechanisms detailed treatment in future work, we believe the level of detail we have included in our paper is sufficient to capture the gist for a general audience.

6) line 123: "High salt": it is only 15 mM, correct? Not sure that is high salt. Also: how much Pi is in cells? Does ADP F-actin exist in cells? (Sorry, I am not an actin expert ...).

We have modified this line to emphasize that our results highlight a specific effect of phosphate (lines 103-104), rather than "high salt" per se.

The estimated Pi concentration in cells ranges from 0.5 – 5 mM [PMID: 21406298], F-actin concentrations range from 100-300 μ M [PMID: 10940259], and the dissociation constant of Pi from F-actin (Kd) is ~1.5 mM [PMID: 3335528]. Back-of-the-envelope equilibrium binding thermodynamic calculations (obviously a gross oversimplification) using these values suggest 25%-80% ADP F-actin in cells (ignoring the small population of ATP-F-actin).

However, to our knowledge the presence of ADP-F-actin in cells has not been directly proven, as this would require a specific probe which discriminates ADP-F-actin from ADP-Pi-F-actin. Nevertheless, a spatial gradient of cofilin localization has been observed at the edge of some cell types [PMID: 10352018], with cofilin depleted proximal to the membrane and enriched distal from the membrane. This is consistent with the prediction of the model in which actin nucleotide state serves as a biochemical marker of filament age as discussed in the introduction.

7) line 126: Not sure two digits needed in cryo-EM resolutions, since FSC estimates are not very precise.

We have updated the resolutions assessments in the main text to only include a single digit after the decimal place.

8) line 130: How were water molecules identified? Were they manually assigned and also checked for chemical plausibility (an old and solved problem in crystallography)?

As noted in the Methods (line 752), waters were initially automatically placed using the phenix.douse program, a recently developed tool for finding plausible solvent peaks in cryo-EM maps. It considers the sphericity of density peaks, their intensity, and geometric considerations (potential for reasonable hydrogen bonding distances and angles), similar to crystallographic tools with which the reviewer may be familiar (e.g. ARP/WARP). They were then manually curated, as noted in the Methods (lines 761-762).

9) line 135: Does the comparison to the Raunser study (Oosterheert et al, bioRxiv 2022) also extend to all/important water molecules?

The Raunser lab's maps and models have not yet been made publicly available (which we do not criticize ours have not yet either). With the information we have (the waters presented in their figures), we believe there are no meaningful discrepancies. However, without a direct comparison of superimposed atomic models, we are not comfortable making any stronger statements than we already have.

10) line 139: Such small RMSD differences will be dominated by the refinement protocol used. Maybe mention this? The number is basically meaningless and does not represent the experiment.

We have modified this sentence to emphasize that this is likely within the error of model building / refinement (line 123).

11) line 146-149: Again, I am not sure this is right: cofilin binding will introduce large conformational differences, but this uses the energy of its binding, and that energy might be in surplus for at least one form. What the energetics or kinetics of such a change are is not necessarily obvious in structures of the starting and final states (which are very similar here in terms of structure (+/- Pi), but it can be the transition state that makes the difference.

Please see our response to point 5.

12) lines 176-177: The interfaces have to be solvated because the subunits (G-actin) need to be soluble in the cytoplasm. This is in contrast to largely hydrophobic interfaces that never or rarely come apart. This is not a surprise or unusual.

While to our knowledge no one has previously shown that inter-subunit interfaces in F-actin are highly solvated, we agree that it is very reasonable based on the physiochemical considerations the reviewer notes. Accordingly, we did not say that this observation was “surprising” or “unexpected”, we simply described what we observed. We do think that this will nevertheless be surprising and striking to many in the actin field, likely changing how they view the filament’s assembly principles.

13) line 192: “Significantly”: how can we be sure this is significant? No estimation of errors given or variances? Line 203: variabilities do seem different, making averages possibly problematic?

We used the term “significantly” because a non-parametric statistical test (the Mann-Whitney test) was consistent with the two distributions being statistically significantly different ($p < 0.0001$), as shown in Fig. 3b and the corresponding legend. Notably, this test does not depend on normality or equivalent variances.

As these distributions have very large numbers of observations, we do note that the p-value is probably not particularly meaningful. Nevertheless, we felt that including it was important to emphasize that the distributions are distinct for readers who expect such a statistical comparison, as this is a major finding of our paper. Furthermore, in the Supplemental Discussion and in the newly included analysis in Extended Data Fig. 4, we demonstrate that the differences in the observed curvature distributions are of the scale that would be expected based on prior studies of F-actin persistence length despite some systematic deviations, a finding that will be of interest to the cytoskeletal biophysics community.

14) lines 204-205: Resolution could be estimated with FSCs against atomic models. 8 Å seems quite low.

We thank the reviewer for this suggestion. To our knowledge, while map-model FSCs are widely employed, they are used to assess overfitting rather than map resolution. Because the model is fit to the map, it is expected that the map-model FSC will show a very high degree of consistency. What is concerning is when this consistency (higher “resolution”) is greater than that calculated from independent half-maps, which would suggest overfitting. Furthermore, the use of map-to-model FSC comparisons for reconstructions derived from cryoDRGN or other variability analysis programs presents issues because the map density is not computed from the raw data and exhibits non-standard gray value distributions, which may confound the measurements.

In response to the reviewer’s comment 17, we now include FSC calculations for the 16-protomer asymmetric reconstructions (Extended Data Fig. 7) which are derived from the same data as the cryoDRGN reconstructions and are visually similar. These all fall within the 6-7 Å range, consistent with our (slightly conservative) interpretation.

Not possible to collect a really big dataset and classify one particular curvature? Ah. I see, this has been done - lines 252-. So, do we still need the cryoDRGN analysis for this study? Why not only use the higher-res data?

We believe that the cryoDRGN analysis and high-resolution analysis of single curvatures give different types of information, both valuable. For instance, the mathematical modelling of the bend-twist relationship, which the reviewer found compelling, requires measuring the helical parameters at different

curvatures. On the other hand, the higher-resolution, single curvature snapshots are appropriate for delineating the internal conformational rearrangements of individual subunits.

15) line 234: “Inter-strand angles also separated by reference strand,” – something wrong with this sentence, I think?

We have clarified this clause. Also, this section of the text, which focuses on detailed analysis of interest to more specialized readers, has been moved to the Supplementary Discussion.

16) lines 239-242: Thank you for pointing this out! (cytomotive switch; G-F transition; flattening is different from bending deformations!

We are glad this point resonated, and thus we have left it in the main text.

17) lines 246-247: Resolutions of those more “traditional” reconstructions similar? Give numbers here?

Apologies for this omission. Resolution assessment of these reconstructions is now included (Extended Data Fig. 7, lines 207-209).

18) lines 252-332: Quite a number of structural and deformation details are listed here, describing very small differences. Some “active”/colourful language used (pressing down etc) that makes looking at the effects disappointing, at least to me.

We appreciate the reviewer’s point that this section is a bit tedious. While we think the active language is useful for visualizing the changes, we concur that the effects are small. We have moved this section of the paper to the Supplementary Discussion, as we believe it will be of intense interest to a subset of specialized readers.

19) lines 345-346: That’s not very clear to me: what does “structural landscape” mean here and how would ABPs work with this? ABPs would then basically just discriminate more-or-less straight/bent filaments? Is that what is meant and is that meaningful in a cell (I don’t know)?

The reviewer’s interpretation is correct. As we state, the structural rearrangements elicited by bending are of a magnitude that ABPs can detect. This is akin to allostery, where force is the allosteric modulator. However, we prefer to avoid the term “allostery”, derived from the Greek *allos* (ἄλλος) *stereos* (στερεός), “other solid (object)”, as it is our opinion that it should be restricted to describing structural transitions modulated by binding transactions. As cofilin and other ABPs have been shown also to be sensitive to forces (including specifically bending for cofilin), as we state in the introduction (lines 72-76), we believe the available evidence suggests this is likely to be meaningful in a cell.

Unfortunately, we are not sure how we can make this particular point clearer within the strict length limitations of Nature, so we are glad the reviewer understood the gist despite some uncertainty in the full interpretation.

20) line 354/364: “Steric boundaries” model - first time this term is introduced here as a conceptual advance. Mention earlier what it is? Needs its own paragraph? Refer to Fig. 6b here.

We have included explicit mention of Figure 6 (now Figure 4f) here (line 272): Unfortunately, due to space restrictions, we cannot have a lengthier discussion of the steric boundaries mechanism which we introduce here. However, we have added some text to the figure panel itself which hopefully clarifies its meaning upon first glance. We plan to give the mechanism a more detailed theoretical treatment, particularly if it has general explanatory power as we anticipate, in future work.

21) Something that could help in the future: if cofilin is added to ADP and ADP-Pi F-actin filaments (under non-saturating conditions), what happens then? Are differences observable by cryo-EM in terms of how many bind to what nucleotide state/curvature?

The reviewer's suggestion is apt. Cofilin acts physiologically under non-saturating conditions, severing at the boundaries between bound and unbound filament regions. One hypothesis suggested by our work is that cofilin may prefer to engage one side of a curving segment, creating conditions likely to impact this activity. We are planning to pursue structures of cofilin bound to bent F-actin under non-saturating conditions, and how this intersects with F-actin nucleotide state, in future work, if others do not beat us to it using the tools we have generated and are freely distributing with the publication of this manuscript.

22) Figures are very well done and very clear!

We are grateful to hear the reviewer found them clear!

23) Figure 1C: For non-experts, label subdomains (two nomenclatures), D-loop, H-loop, nucleotide binding pocket etc

We have labelled Figure 1c. The H-plug is not readily visible in this figure panel, so this particular requested label was omitted.

24) Figure 4e: Why is there so much scatter in the AU units that seems non-random?

For the purposes of comparing total nucleotide cleft strain and solvent-accessible pocket volume we measured these values for each protomer and pooled them based on nucleotide state and bending state. Each protomer in the bent models experiences a unique lattice environment, and we observe that the protomers deform at the subdomain level according to their position in the lattice. Therefore, we might expect the distribution of these metrics to be a complicated function of lattice position. Furthermore, the sparse sampling from such a short segment unit makes it difficult to assess finer details within the distributions. For the purposes of the comparison between nucleotide state and bending conditions, we found that the differences were significant even with these confounding factors.

For the solvent accessible volume measurements, the measured volumes features some apparent discretization, likely due to the discrete probe size used for the measurement.

25) Figure 6a: How do the ABPs mentioned bind precisely? Supp figure might be useful to understand how such small differences could lead to differential effects?

We have now included Extended Data Figure 11, which shows each relevant ABP's binding site directly. For conciseness, we have also moved former panel Fig. 6A to this figure.

Also: the "ABP binding box" includes two nucleotide binding pockets. Can these ABPs really not sense nucleotide much more directly? (Sorry, don't know).

We appreciate the reviewer pointing out this ambiguity. We have now included a clause in the introduction (line 84) explicitly stating that the nucleotide cleft is buried in F-actin, and ABP's thus cannot directly access / contact the nucleotide to "read" F-actin's nucleotide state.

25) Figure 6b is important for the manuscript. Move its reference out of discussion / have a chapter summarising the model/mechanism?

26) Figure 6b: a brief explanation what is depicted in 6b would have helped me a lot to understand quicker. 6b is not referenced in the text, I think?

Here responding to both points 25 and 26: as noted in our response to point 20, we do not have the space to include an expanded discussion of the steric boundaries mechanism here. We have now included an explicit reference to Figure 6b (now Figure 4f) in the Discussion text, which hopefully makes the connection between the figure and the text clearer. We have also included text on the figure panel itself, which we hope will aid comprehension when a reader first looks at the figure panel.

27) Supp Table 1: Why did ADP-P_i need twice as many particles to reach a lower resolution?

While we cannot give a definitive answer, we believe there are two potential factors at play:

a) The ADP dataset was collected with stage translations, without using image-beam shift. Thus, the images contain fewer electron optical aberrations than the ADP-P_i dataset, which was collected with image-beam shift. While these aberrations can be corrected computationally, this may be imperfect, potentially necessitating more particles to achieve the same resolution.

b) The ADP specimen had lower filament density (something which is difficult to perfectly control), and likely featured thinner ice. We did not explicitly measure this, but it is apparent in inspecting the images. We believe b is likely the major culprit, but we have not systematically explored this. As we are making our raw data available on EMPIAR, methods developers who are interested in these questions will be able to address them with our data, if desired. We believe this likely to be of value to the community, as the majority of cryo-EM reconstructions thus far reported with sufficient resolution to visualize water molecules have been from a small number of test specimens (notably apoferritin).

Referee #2 (Remarks to the Author):

In this work, Reynolds and coworkers use cryo-electron microscopy to solve new structures of filaments actin in ADP and ADP+free phosphate states. These structures allow them to detail possible roles for hydrogen bonded solvent in mediating interactions between actin subunits. Then, developing and training a machine learning algorithm to classify actin filament segments by curvature, the authors solve the structure of actin filaments for various amounts of bending. These structures contain as yet previously un-measured changes subunit structure (nucleotide dependent) that have implications for affinity of actin-binding proteins and, overall, the mechanical regulation of the actin cytoskeleton. The latter results, and in particular, the methodology leading to them, are completely novel to my understanding. Therefore, the results of this manuscript have the potential to make a large impact both on our understanding of stressed actin filaments, as well as the way in which researchers process their EM data. Below, I have a few comments and questions that I hope will help improve the manuscript.

We thank the reviewer for their recognition of the significance of our study and the constructive / highly useful comments, which we believe have helped us substantially strengthen the manuscript.

1) Because a substantial part of the manuscript is about the role of bound waters, it would be helpful to compare these results to prior studies discussing the role of water in the nucleotide pocket. In particular, this paper from Voth (10.1016/j.jmb.2011.07.068) is a detailed study of this topic. But, perhaps that study is now out of date based on these new data, which would be very helpful to know. It would also be good to know how these new data compare to detailed studies on the actual hydrolysis process and the final state, if possible (10.1021/acs.jctc.7b00077).

We have now included a brief comparison with that provided in the figures of 10.1016/j.jmb.2011.07.068 in the new Supplementary discussion. Unfortunately, the simulation results do not appear to be publicly available, and thus we cannot provide a detailed assessment. This is particularly true for 10.1021/acs.jctc.7b00077, where the figures do not indicate specific distances or other details of the post hydrolysis state.

During the course of this analysis and that related to point 3 below, we also made some slight improvements to the water positioning in the high-resolution models, and thus the statistics in Extended Data Table 1 have been updated (as have the models deposited with the PDB). These did not impact any of the waters already displayed, and the figures and movies have not been changed.

2) I'm unclear as to the resolving of D-loop conformations in the various structures. The text and supplemental figure 2 discuss the variability in the D-loop. However, Fig1C to me looks like the D-loops

are superimposed almost exactly. Is that because it was modeled in using the same dihedrals despite uncertainty in the electron density? Perhaps this could be clarified further.

This is a case where analyzing the data in different ways produces different map features. In our high-resolution maps, every protomer position is averaged together when helical symmetry is applied. After this procedure, one D-loop conformation (the “Out” conformation) dominates, and this is the density path into which we built our atomic models. In a recent paper (PMID: 35857845) using symmetry expansion and focused classification we showed that ~80% of subunits feature the D-loop in this conformation in ADP F-actin, consistent with this result.

In our asymmetric reconstructions (both bent and straight), there is no overlap between the segments being averaged, and thus each subunit position will represent the average of distinct protomer subpopulations, which may deviate from the global average. Notably, these protomers (particularly in the straight controls) represent samples from the same population as those used to generate the high-resolution reconstruction.

As we extensively discussed this issue in our recent paper (PMID: 35857846, ref. 41) specifically focused on D-loop dynamics, we have elected not to engage in extensive discussion here beyond that we have already included (now in the Supplementary Discussion).

3) For water bridging interactions between subunits, I'm curious if the authors could make a comparison and statement about the possible role of divalent cations in bridging the subunits at these locations, as described in 2-3 papers by Enrique De La Cruz and coworkers. Would inclusion of specifically bound cations at these locations be able to be accommodated in these new structures or would it require substantially different water mediated interactions?

We have included a brief discussion of this in the Supplementary Discussion. The summary is that we observe two density peaks in the “polymerization site” at the intersubunit longitudinal interface that we have modelled as water molecules but which could feasibly be divalent cations. We do not see any such density peaks in the “stiffness” site, but as this site falls within subdomain 2, the lowest resolution region of our maps, this may simply be a resolution limitation. Our current data do not have the capacity to discriminate cations from water molecules (with the exception of the highly coordinated magnesium in the nucleotide cleft).

4) Bent actin filaments

a) It would be helpful to give the curvature values in Fig 3c,d, and in other locations where curvature is illustrated (e.g. S4a,b)

We have now indicated the curvature in figure panels Fig. 3c, Extended Data Fig. 5a, and Extended Data Fig. 7a. We did not include them in Fig. 4a, as we are less confident in the precision of our curvature measurements for the short 7-protomer F-actin segments. However, we estimate that they are close to the mean curvature for the subpopulation of bent particles in each dataset (curvature measured from models was $3.9 \mu\text{m}^{-1}$ for ADP-F-actin and $3.1 \mu\text{m}^{-1}$ for ADP-P_i-F-actin). Similarly, we did not include them for the stitched volumes in Fig. 3d and Extended Data Fig. 5b, as these are not real experimental volumes as described in point b below. Nevertheless, we believe the visual comparison with the included scale bar will be helpful for our cell biologist readership.

b) How long are the filaments used to compute the structures of curved segments? It seems for solving the structure, always the same number of subunits/protomers are used (16), but is this structure then coming from many adjacent/overlapping regions along a curved region? 16 promoters is I guess $2.6 \times 16 \sim 42\text{nm}$ long [by the way, line 924 says 56 angstroms which is the longitudinal spacing between actin subunits as stated on line 927, but the text says this is the length of one promoter. I'm not sure if that's a typo or not].

The reviewer is correct: for the cryoDRGN analysis of bent segments, we used 16-protomer long segments. These do not feature overlap in the final reconstructions, one reason why they are somewhat

lower resolution (segment number becomes limiting). For viewing features that continuously deviate from symmetry along the region being analyzed, overlap is a confounder we wish to avoid.

We realize that in the modelling literature, the “protomer length” is often referred to as ~28 Angstroms, the axial rise. This is the amount an actin filament is extended when a new subunit is added, which we believe is how this term was introduced in the context of Brownian ratchet models. However, from a structural perspective a physical actin subunit occupies 56 Angstroms along the filament axis, which is how we construe its “length”.

The highly curved segments appear to be on the order of 50 nm long in Fig 3a, but the structures illustrating curvature in Fig 3d are perhaps 300nm long, so I'm not sure which is representative of most of the data.

The reconstructions shown in Figure 3d are end-to-end stitches of multiple copies of each cryoDRGN volume, an operation we performed to emphasize their curvature on a cellular scale. This is admittedly a somewhat artistic exercise, and we did not use these stitched volumes for any detailed analysis, but we believe this will resonate with cell biologist readers. We now explicitly explain this in the text (lines 173-175); we inadvertently did not include this in our initial submission, which we realize was confusing.

c) Related, how long is the model filament used for training in terms of number of actin subunits? I presume a fixed length filament is bent in order to generate the training data.

Since 2d images of synthetic bent filaments are used based on these data, do these 2d images always have the filament centered in the middle of the image, or could it have just a small piece of the filament in the synthetic imaging area?

The reviewer is correct. A model filament of fixed length (35 subunits), substantially longer than the box size of the projections ultimately used for training, was bent. This model filament is then randomly rotated and translated throughout the box: it is not always centered. Thus, in some training images only a small segment will be in the synthetic projection. The last paragraph in the Methods section “Synthetic dataset generation” has been updated to make this more explicit (lines 568-585).

Nevertheless, while we believe this is important for accurate picking, as the network is highly likely to encounter image regions fulfilling this condition in real data, it does not impact the curvature measurements reported in the paper. The reported measurements are derived from image segments after picking, which were selected based on the semantic segmentation map of entire micrographs. These segments will all be well-centered, and therefore they will all feature a nearly identical (varying slightly depending on their curvature) filament contour length / number of protomers in the segmented area.

d) Finally and related, in figure 3b, what is the length of actin subunits that are histogrammed here. Are they all the same length at fixed radius of curvature? This relates to my next point.

All of the filament segments have nearly identical length as described above. A more bent filament will have a slightly longer contour length sampled due to the discrete windowing by the particle box size, but this is likely negligible for the reviewer's question.

e) A question I still have reading the manuscript is how likely are these curvatures to arise by thermal fluctuations, and how likely are they with a particular amount of shear force. I believe the paper would be strengthened with some back of the envelope calculations to this end.

This is a great suggestion: we really appreciate the thought the reviewer put into this point!

As a thought experiment, I considered what would give rise to the distribution in Figure 3b. Roughly, the energy to bend a filament of length L to a curvature κ should be something like $\frac{1}{2} k_{\text{bend}} * L * \kappa^2 = \frac{1}{2} kT * L_p * L * \kappa^2$ where L_p is the persistence length (with perhaps a factor of two

wrong because it is bent in an arc and not a circle). The distribution of bends without force is Boltzmann distributed, $P(E) \sim e^{-E/kT}$, so in terms of bending, a half gaussian, $P(\kappa) \sim e^{-1/2 L_p L \kappa^2}$ (and then normalized). If I assume a persistence length of 10 microns and a length of 50 nm, I get a distribution not unlike that in figure 3b. However, I don't know the proper length L to use as in my previous question. This also assumes the filament behaves like a single elastic rod and not a discrete molecular system. Nevertheless, I think this distribution could be computed, and the data in fig 3b could actually be used to compute a p-value of whether this comes from thermal fluctuations as the null hypothesis.

Interestingly, the expected distribution depends strongly on length of filament when the x-axis is curvature, so I don't know if my assumptions are correct here.

There may also be a way to estimate what force would produce a distribution similar to what is seen, which probably does extend to larger curvatures than expected based on the previous argument, and this would be very helpful to know.

We have pursued the calculations the reviewer suggested, now included as Extended Data Fig. 4, and discussed in the main text (lines 160-163) and Supplementary Discussion. With the caveats that the reviewer notes (e.g. the validity of assuming an elastic rod vs. the reality of a discrete molecular system at the nanometer length scale being examined), we find that our curvature distributions are almost but not quite those expected from thermal fluctuations. Notably, the residuals are clearly non-random, even when we adjust the model to include a fitted multiplicative parameter in the energy term (effectively adjusting for persistence length inaccuracies), suggesting an increased proportion of highly curved segments. Fortunately, this long tail is enriched in the set of particles with curvatures we had initially selected based on an arbitrary cutoff of 2.5 microns. Furthermore, rough calculations of the reconstructed cryoDRGN maps indicated that they had associated bending energies in the range of ~ 1 to $7 k_B T$, consistent with bending deformations of this scale being associated with active bending.

In summary, we believe that the distribution represents a mixture of thermal fluctuations (major contributor) and shear forces (minor contribution), but we have focused our structural analysis of bent actin on a subpopulation enriched for filaments experiencing shear forces. We believe this analysis substantially enhances the paper.

Additionally, the reviewer's efforts made us realize that the raw distributions of curvature values we measured may be of interest to the modelling community, and thus we are making them publically available through Zenodo as indicated in the Data Availability statement.

5) Additional data:

As far as I can tell, the current manuscript doesn't have the pDBs of the new models included (just the validation for submission), and the GitHub link for the software is not available yet. It would be nice to be able to quickly review these. Additionally, I suggest the authors include the synthetic training set for the neural network models of actin filaments bent by different curvatures, as that may be helpful for future researchers.

We apologize that this was not included in the initial submission. The raw structural data (maps and models deposited with the PDB) were too large to be uploaded through *Nature's* submission portal. We have now transmitted them to the journal through Google drive to be made available to the reviewer.

Furthermore, we have made pooled curvature measurements, the training data, trained neural networks for micrograph segmentation and particle picking, input data to train cryoDRGN neural networks, trained cryoDRGN neural networks, cryoDRGN maps and fit models, and cofilin TIRF data and code available through Zenodo (already publicly available through the provided DOIs). We have also made the GitHub repository with the code publicly available. The raw cryo-EM micrographs will also be made available through EMPIAR upon acceptance and publication, along with the high-resolution maps / models through the EMDB / PDB. All accession codes / DOI URLs are provided in the Data Availability and Code Availability sections in the main text.

Reviewer Reports on the First Revision:

Referees' comments:

Referee #1 (Remarks to the Author):

Second round review: Reynolds et al., "Bending forces and nucleotide state jointly regulate F-actin structure", Nature manuscript # 2022-06-08690A (2022)

The authors have done an excellent job in refining, shortening and improving their manuscript. The revised manuscript concentrates on the main outcomes that provide a detailed and convincing description and analysis of F-actin conformations in different nucleotide and bending states. There is significant novelty in their work, both in terms of findings but also in the methods they used and developed.

My major criticism during the first round of reviewing was that the prediction that ABPs may detect these changes as a proxy for nucleotide state was over-emphasised - this has now been rectified. For example, the title and abstract are now a much better fit for what the work is about, at least in my view.

Removing tedious descriptions of small effects has also helped to focus readers on what matters – but it is good that details have been retained in the supplementary discussion, for example.

I still think that the language used tends to over-complicate things in some places, for example around the "steric boundaries" model (Figure 6), but I can forgive that.

All my specific comments and questions were answered with precision and care, and acted upon.

It has been a pleasure to help this outstanding work getting published.

Referee #2 (Remarks to the Author):

I feel the authors have sufficiently addressed the comments of the referees (and I especially appreciate them making all of that data available through Zenodo and code/methods in github). I feel that it is suitable for publication.

Minor comment:

In the extended discussion, the Boltzmann weights appear to be missing a minus sign in the exponentials.

Author Rebuttals to First Revision:

Referee #1 (Remarks to the Author):

Second round review: Reynolds et al., "Bending forces and nucleotide state jointly regulate F-actin structure", Nature manuscript # 2022-06-08690A (2022)

The authors have done an excellent job in refining, shortening and improving their manuscript. The revised manuscript concentrates on the main outcomes that provide a detailed and convincing description and analysis of F-actin conformations in different nucleotide and bending states. There is significant novelty in their work, both in terms of findings but also in the methods they used and developed.

My major criticism during the first round of reviewing was that the prediction that ABPs may detect these changes as a proxy for nucleotide state was over-emphasised - this has now been rectified. For example, the title and abstract are now a much better fit for what the work is about, at least in my view.

Removing tedious descriptions of small effects has also helped to focus readers on what matters – but it is good that details have been retained in the supplementary discussion, for example.

I still think that the language used tends to over-complicate things in some places, for example around the "steric boundaries" model (Figure 6), but I can forgive that.

All my specific comments and questions were answered with precision and care, and acted upon.

It has been a pleasure to help this outstanding work getting published.

We thank the reviewer for their highly constructive critiques and recognition of the significance of our work. We are pleased that the reviewer supports publication of our work in *Nature*.

Referee #2 (Remarks to the Author):

I feel the authors have sufficiently addressed the comments of the referees (and I especially appreciate them making all of that data available through Zenodo and code/methods in github). I feel that it is suitable for publication.

We thank the reviewer for their highly constructive critiques and recognition of the significance of our work. We are pleased that the reviewer supports publication of our work in *Nature*.

Minor comment:

In the extended discussion, the Boltzmann weights appear to be missing a minus sign in the exponentials.

Thanks for catching this: we have added minus signs to the relevant equations.